# STING induces HOIP-mediated synthesis of M1 ubiquitin chains to stimulate NF-κB signaling

Tara D Fischer [1 ✉], Eric N Bunker [1], Peng-Peng Zhu [1], François Le Guerroué[1,5], Mahan Hadjian[1], Eunice Dominguez-Martin [1], Francesco Scavone [2,6], Robert Cohen [2], Tingting Yao[2], Yan Wang [3], Achim Werner [4] & Richard J Youle [1 ✉]

## Abstract

STING activation by cyclic dinucleotides induces IRF3- and NF-κB-mediated gene expression in mammals, as well as lipidation of LC3B at Golgi-related membranes. While mechanisms of the IRF3 response are well understood, the mechanisms of NF-κB activation via STING remain unclear. We report here that STING activation induces linear/M1-linked ubiquitin chain (M1-Ub) formation and recruitment of the LUBAC E3 ligase, HOIP, to LC3B-associated Golgi membranes where ubiquitin is also localized. Loss of HOIP prevents formation of M1-Ub chains and reduces STING-induced NF-κB and IRF3 signaling in human THP1 monocytes and mouse bone marrow-derived macrophages, without affecting STING activation. STING-induced LC3B lipidation is not required for M1-Ub chain formation or for immune-related gene expression, but the recently reported STING function in neutralizing Golgi pH may be involved. Thus, LUBAC synthesis of M1-linked ubiquitin chains mediates STING-induced innate immune signaling.

**Keywords** Golgi; Innate Immunity; NFkB; LUBAC; LC3B
**Subject Categories** Immunology; Organelles; Post-translational Modifications & Proteolysis

## Introduction

The evolutionarily conserved cGAS-STING pathway initiates potent innate immune responses through several signaling cascades following the detection of double-stranded DNA (dsDNA) in the cytoplasm of cells (Chen et al, 2016; Hopfner and Hornung, 2020; Ahn and Barber, 2019). In mammals, cGAS synthesis of the cyclic dinucleotide 2′3′ cyclic GMP-AMP (cGAMP) and its binding to STING at ER membranes induces the trafficking of STING from the ER to the Golgi apparatus. Following trafficking through Golgi compartments, and prior to its degradation in the lysosome, STING

initiates a broad transcriptional program of type I interferons mediated by its interaction with the kinase TBK1 and the transcription factor IRF3. At Golgi membranes, active STING also induces the lipidation of the ubiquitin-like protein, LC3B, through mechanisms that are distinct from the known role of LC3B lipidation in autophagosome formation (Gui et al, 2019; Fischer et al, 2020; Mizushima, 2020). Activation of cGAS-STING also induces the transcription of NFκB-related genes through poorly understood mechanisms (Ishikawa and Barber, 2008; Abe and Barber, 2014; de Oliveira Mann et al, 2019; Balka et al, 2020; Yum et al, 2021). Although all of these downstream signaling events mediated by STING activation have been reported to play a role in antiviral defense (Gui et al, 2019; Ishikawa and Barber, 2008; Yum et al, 2021; Zhong et al, 2008; Ishikawa et al, 2009), whether and how they are mechanistically related is unclear. Here, we report that ubiquitin robustly co-localizes with activated STING and LC3B at Golgi membranes. As ubiquitylation is important in both autophagy-related and innate immune signaling, we sought to determine whether these ubiquitylation events play a role in STING-mediated innate immune responses.

## STING activation induces M1- and K63-ubiquitin chain formation

Activation of STING and its trafficking to the perinuclear region of cells induces clusters of small LC3B positive vesicles near the Golgi apparatus (hereafter referred to as LC3B foci) (Gui et al, 2019; Fischer et al, 2020; Prabakaran et al, 2018). Immunostaining for ubiquitin (Ub) after treatment with the STING ligand, 2′3′ cGAMP, or the STING agonist, diABZI, in HeLa cells stably expressing untagged STING (HeLa[STING]), at low levels, and mEGFP-LC3B showed that >80% of cells present Ub positive foci that co-localize with mEGFP-LC3B and a subset of STING punctae in the perinuclear region (Figs. 1A,B and EV1).

Ubiquitin can be conjugated to other ubiquitin molecules at any of the seven lysine residues and to the N-terminal methionine (M1), forming unique poly-ubiquitin chains that mediate specific downstream signaling pathways (Komander and Rape, 2012). To

[1]Biochemistry Section, Surgical Neurology Branch, National Institute of Neurological Disorders and Stroke, National Institutes of Health, Bethesda, MD, USA. [2]Department of Biochemistry and Molecular Biology, Colorado State University, Fort Collins, CO, USA. [3]Mass Spectrometry, National Institute of Dental and Craniofacial Research, National Institutes of Health, Bethesda, MD, USA. [4]Stem Cell Biochemistry Unit, National Institute of Dental and Craniofacial Research, National Institutes of Health, Bethesda, MD, USA. [5]Present address: Single Cell Biomarkers UTechS, Institut Pasteur, Université Paris Cité, Paris, France. [6]Present address: Department of Biology, Stanford University, Stanford, CA, USA. ✉E-mail: fischertad@nih.gov; youler@ninds.nih.gov

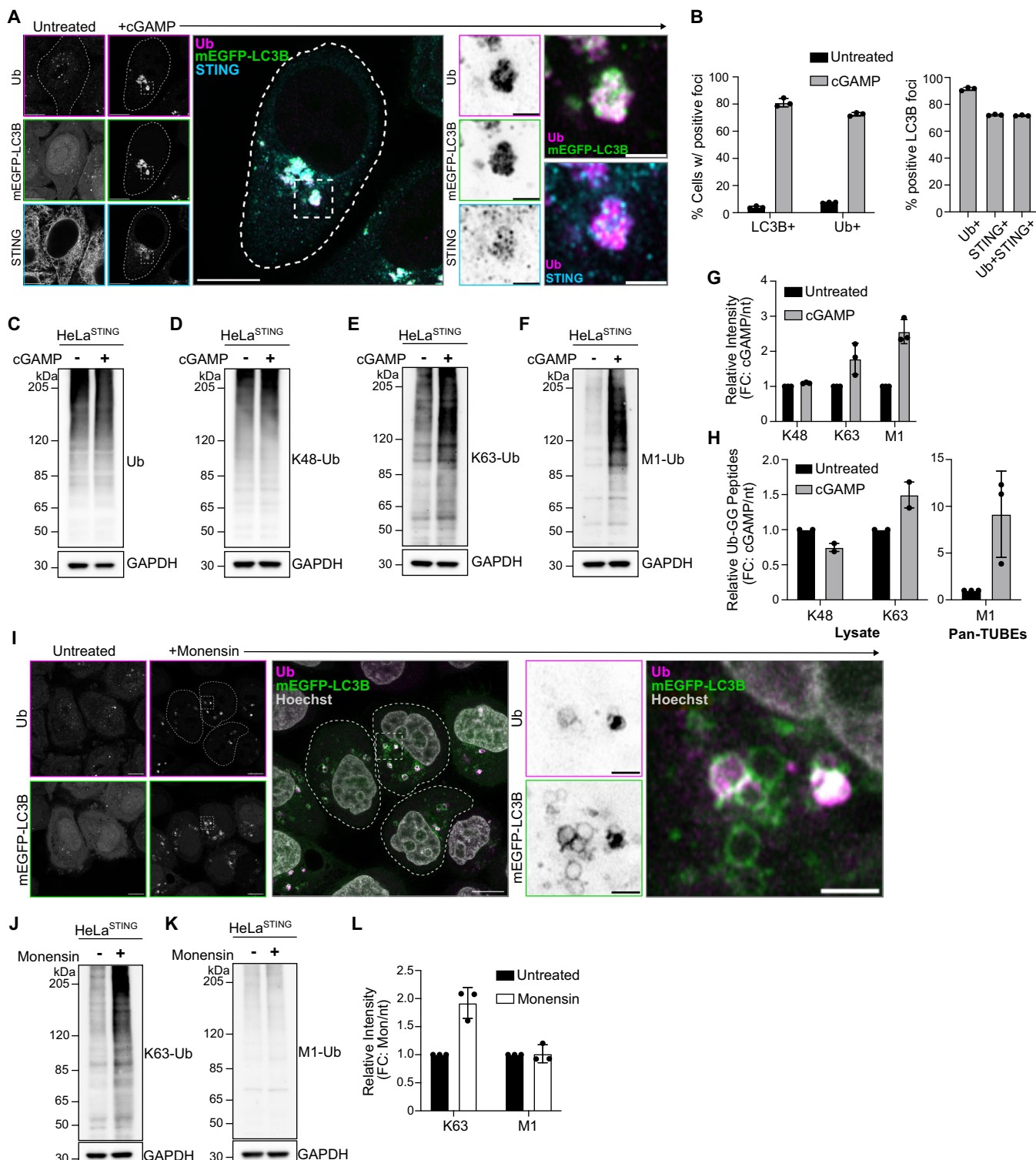

detect which poly-ubiquitin chain species are localized at LC3B-foci, we used linkage-specific ubiquitin antibodies to immunostain for K48- and K63-Ub chains (Fig. EV1). Robust K63-Ub positive foci were detected in 67% of cells, with over 80% of mEGFP-LC3B foci staining positively for K63-Ub (Fig. EV1). However, no K48-Ub foci were detected in cells, and no mEGFP-LC3B foci stained

positively for K48-Ub (Fig. EV1). K63-Ub co-localization with LC3B following STING activation by cGAMP or diABZI in live cells was also detected using a fluorescently tagged probe that selectively binds K63-Ub chains (Sims et al, 2012) (Fig. EV1; Movie EV1). Consistently, the detection of linkage-specific poly-ubiquitin chains by immunoblotting showed a cGAMP-induced

◄ **Figure 1. STING activation induces M1 and K63-ubiquitin chain formation.**

(A) Representative Airyscan-processed confocal images of wild-type (WT) HeLa cells stably expressing BFP-P2A-STING (HeLa^STING) and mEGFP-LC3B (green) treated with 60 μg/mL of cGAMP for 8 h prior to PFA-fixation and immunostaining with antibodies raised against mono- and poly-ubiquitin chains (Ub; magenta) and STING (cyan). Scale bar = 10 and 2 μm (inset). Imaging was replicated in >3 independent experiments. (B) Quantification of the percentage (%) of cells positive for mEGFP-LC3B foci and immunostained Ub foci (left panel), and the percentage (%) of mEGFP-LC3B foci with overlapping immunolabeled signal for Ub, STING, or both (right panel) from experiments represented in Fig. EV1A, and similar conditions to Fig. 1A. Mean ± s.d. from *n* = 3 replicates analyzed in the same experiment. Imaging was replicated in three independent experiments. (C–F) Representative immunoblots of indicated proteins detected in HeLa^STING cell lysates prepared after treatment with 120 μg/mL cGAMP for 8 h. Immunoblotting was replicated in three independent experiments. (G) Quantification of immunoblots in (D–F). Values represent the relative intensity of cGAMP treated to untreated lanes. Error bars represent the mean ± s.d. of three independent experiments. (H) Quantification of ubiquitin-GG linked peptides from lysate (left) and Pan-TUBE (tandem-ubiquitin-binding entities) enriched samples (right) identified by targeted LC-MS/MS. HeLa^STING cells stably expressing mEGFP-LC3B were treated with 120 μg/mL of cGAMP for 8 h prior to cell lysis, Pan-TUBE enrichment, and LC-MS/MS analysis. Values represent the relative amount of peptides detected in cGAMP-treated cells over untreated cells. (Lysates) Mean ± s.d. from *n* = 2 independent experiments with 3–5 technical replicates each; (Pan-TUBEs) Mean ± s.d. from *n* = 1 of the same cell lysates analyzed in the lysate panel with three technical replicates. (I) Representative Airyscan-processed confocal images of HeLa^STING; mEGFP-LC3B (green) cells treated with 100 μM Monensin for 1 h prior to PFA-fixation and immunostaining with antibodies raised against mono- and poly-ubiquitin chains (Ub; magenta). Scale bar = 10 and 2 μm (inset). Imaging was replicated in two independent experiments. (J, K) Representative immunoblots for indicated linkage-specific ubiquitin chains in HeLa^STING cell lysates prepared after treatment with 100 μM Monensin for 1 h. Immunoblotting was replicated in three independent experiments. (L) Quantification of immunoblots in (J, K). Values represent the relative intensity of Monensin treated to untreated lanes. Error bars represent the mean ± s.d. of three independent experiments. Source data are available online for this figure.

increase in K63-Ub, but not K48-linked or total-Ub (Fig. 1C–E,G). By immunoblotting, we also probed for linear/M1-linked ubiquitin chains (hereafter referred to as M1-Ub) following STING activation. Surprisingly, cGAMP activation of STING induced a robust increase in M1-Ub chains (Fig. 1F,G). To confirm the distinct poly-ubiquitin chains independently of antibody detection, we used mass spectrometry to quantitatively identify GG-K/M linked peptides in cells following STING activation (Fig. 1H). Detection of GG-K/M linked peptide abundances in lysates from untreated and cGAMP-treated cells showed a small increase in GG-K(63) peptides, and no change in GG-K(48) peptides (Fig. 1H). M1-Ub chains were detected after enrichment for ubiquitin using Pan-TUBE (tandem-ubiquitin-binding entities) pulldown, showing an eight-fold average increase in GG-M(1) peptide abundance with cGAMP treatment compared to the untreated control (Fig. 1H). Collectively, these data demonstrate that STING activation induces robust ubiquitin co-localization with LC3B foci in the perinuclear region, that prior work showed is the Golgi apparatus (Fischer et al, 2020; Gui et al, 2019; Liu et al, 2023; Xun et al, 2023). STING activation also induces the synthesis of M1 ubiquitin chains, in addition to K63-ubiquitin chains.

LC3B lipidation at acidic organelles can be induced by a variety of stimuli, including lysosomotropic agents, such as Monensin (also referred to as CASM) (Jacquin et al, 2017). To determine whether ubiquitin co-localization with LC3B is a general feature of LC3B lipidation at acidic organelles, we immunostained for ubiquitin following Monensin treatment (Fig. 1I). Ubiquitin was detected at some, but not all, Monensin-induced LC3B+ vesicles (Fig. 1I). Immunoblotting for K63- and M1-Ub chains following Monensin treatment showed an increase in K63-Ub chains, but not M1-Ub chains (Fig. 1J–L), indicating that M1-Ub chain formation may be selectively associated with STING-mediated LC3B lipidation at Golgi membranes.

## HOIP is required for STING activation-induced M1-Ub chain formation in HeLa and immune cell lines

While many E3 ligases generate K63-ubiquitin chains, M1 ubiquitin chains are only known to be formed by the E3 ligase

HOIP, a component of the Linear Ubiquitin Assembly Complex (LUBAC) (Kirisako et al, 2006; Sasaki and Iwai, 2015). Stable overexpression of mEGFP tagged HOIP in HeLa^STING cells, also stably expressing mScarletI-LC3B, showed a cytosolic localization in untreated cells (Fig. EV2). cGAMP activation of STING induced a clustering of mEGFP-HOIP in the perinuclear region that co-localized with mScarletI-LC3B foci and STING (Figs. 2A,B and EV2). The localization of mEGFP-HOIP was further resolved following a brief saponin extraction prior to fixation (Figs. 2A and EV2), indicating a stable association with LC3B-labeled perinuclear Golgi structures. The co-localization of HOIP and LC3B foci suggests that the ubiquitin species detected there may be M1-Ub chains, in addition to K63-Ub.

To determine whether HOIP is required for M1-Ub chain formation induced by STING activation, we generated a HOIP knockout HeLa cell line (HOIPKO HeLa^STING). M1-Ub chains following STING activation by either cGAMP (Figs. 2C,D, and EV2) or diABZI (Fig. EV2) were eliminated in HOIPKO HeLa^STING cells, while K63-Ub chains were unaffected (Fig. EV2). Stable reconstitution of HOIP in HOIPKO HeLa^STING cells rescued M1-Ub chain formation following cGAMP activation of STING, confirming that HOIP is required for STING-induced M1-Ub chain formation (Fig. 2C,D). Stable overexpression of the M1 ubiquitin-specific deubiquitylase OTULIN in HeLa^STING cells, which has been shown to counteract HOIP's E3 ligase activity (Fiil et al, 2013; Keusekotten et al, 2013), also eliminated M1-Ub chains, but not K63-Ub chains, following STING activation (Fig. EV2).

STING activation, indicated by LC3B lipidation and phosphorylation of STING at serine 366 following either cGAMP or diABZI treatment, was largely unaffected in HOIPKO HeLa^STING cells over time (Fig. EV2). OTULIN overexpression also caused no difference in LC3B lipidation or STING phosphorylation following cGAMP treatment (Fig. EV2). While there appeared to be a modest disruption of STING degradation at later timepoints in HOIPKO HeLa^STING cells following either cGAMP or diABZI treatment (Fig. EV2), this effect was not rescued with HOIP reconstitution (Fig. 2C), and not observed with endogenous STING in HOIPKO HeLa cells (Fig. EV2), or in OTULIN overexpressing HeLa^STING cells (Fig. EV2). We further sought to determine whether STING

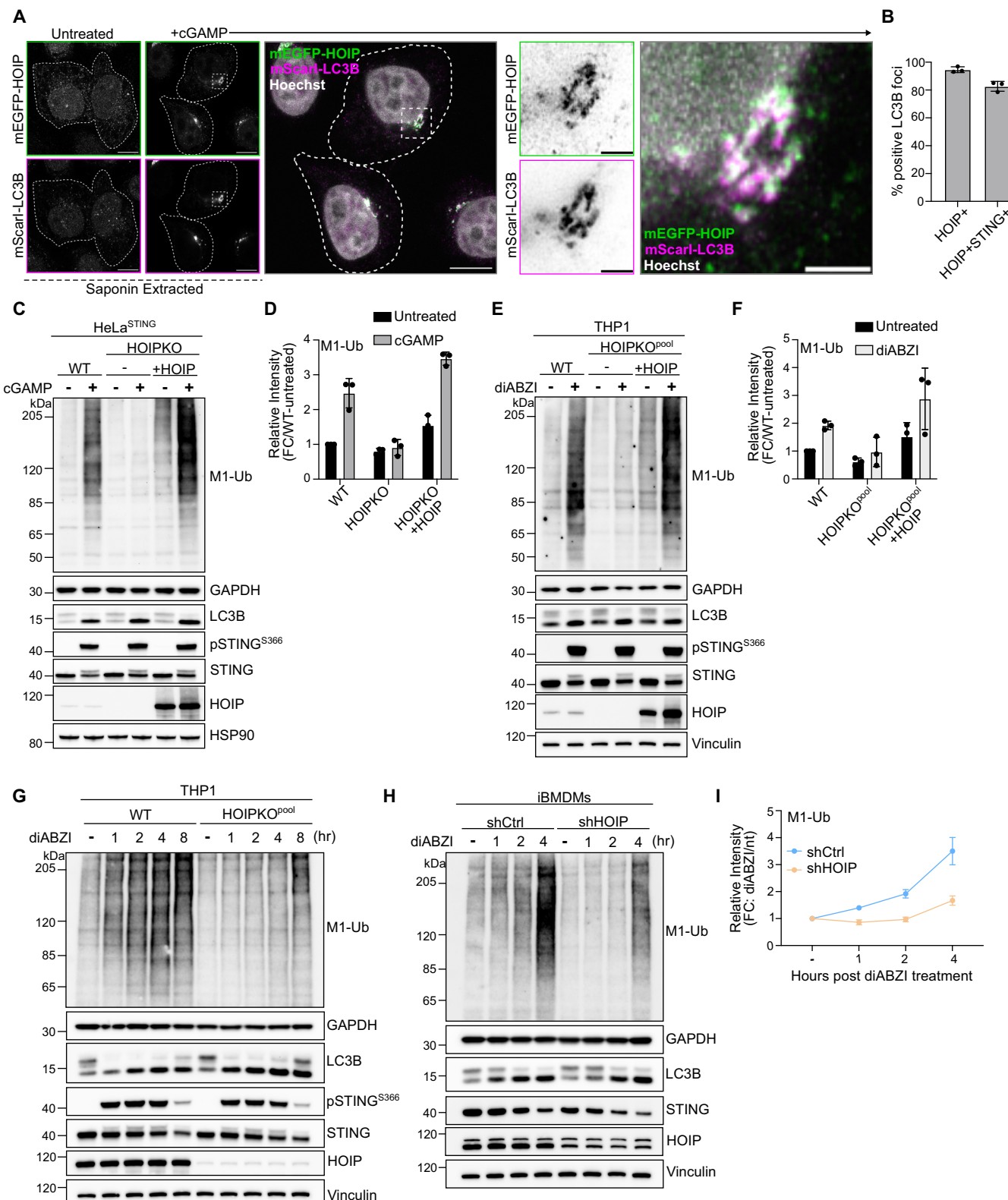

induces HOIP synthesis of M1-Ub chains and the effect of HOIP loss on STING signaling in well-established, immune-relevant cell models with robust endogenous STING expression. Activation of endogenous STING with either cGAMP or diABZI in the human

monocyte, THP1, cell line (Figs. 2E–G and EV2), and diABZI in an immortalized mouse bone marrow-derived macrophage cell line (Fig. 2H,I), induced formation of M1-Ub chains. Knockout of HOIP in THP1 cells (HOIPKO^pool) (Figs. 2E–G and EV2) or

◀ **Figure 2. HOIP mediates STING activation-induced M1 ubiquitin chain formation.**

(A) Representative Airyscan-processed confocal images of HeLa^STING cells stably expressing mScarletI-LC3B (magenta) and mEGFP-HOIP (green) treated with 120 μg/mL of cGAMP for 8 h prior to saponin extraction and PFA-fixation. Scale bar = 10 and 2 μm (inset). Imaging was replicated in three independent experiments. Representative images from non-saponin extracted cells are in Fig. EV2A. (B) Quantification of the percentage (%) of mEGFP-LC3B foci with overlapping immunolabeled signal for HOIP, or HOIP and STING, from experiments represented in Fig. EV2B, and corresponding conditions to Fig. 2A. Mean ± s.d. from n = 3 replicates analyzed in the same experiment. Imaging was replicated in three independent experiments. Representative images are in Fig. EV2B. (C) Representative immunoblots of indicated proteins detected in HeLa^STING cell lysates from WT, HOIPKO, and HOIPKO stably expressing untagged HOIP prepared following treatment with 120 μg/mL cGAMP for 8 h. Immunoblotting was replicated in three independent experiments. (D) Quantification of M1-Ub in (C). Values represent the relative intensity of corresponding lanes to WT untreated lanes. Error bars represent the mean ± s.d. of three independent experiments. (E) Representative immunoblots of indicated proteins detected in THP1 cell lysates from WT, HOIPKO^pool, and HOIPKO^pool stably expressing untagged HOIP prepared following treatment with 1 μM diABZI for 4 h. Immunoblotting was replicated in three independent experiments. (F) Quantification of M1-Ub in (E). Values represent the relative intensity of corresponding lanes to WT untreated lanes. Error bars represent the mean ± s.d. of three independent experiments. (G) Representative immunoblots of indicated proteins detected in THP1 cell lysates from WT and HOIPKO^pool cells prepared following treatment with 1 μM diABZI for 1, 2, 4, and 8 h. Immunoblotting was replicated in three independent experiments. (H) Representative immunoblots of indicated proteins detected in iBMDM cell lysates from shCtrl and shHOIP cells prepared following treatment with 0.2 μM diABZI for 1, 2, and 4 h. Immunoblotting was replicated in three independent experiments. (I) Quantification of M1-Ub in (H). Values represent the relative intensity of corresponding lanes to untreated lanes per cell line. Error bars represent the mean ± s.d. of three independent experiments. Source data are available online for this figure.

knockdown of HOIP in iBMDMs by shRNA (shHOIP) (Fig. 2H,I) substantially reduced the detection of M1-Ub chains following STING activation. M1-Ub chain formation following STING activation was rescued by stable reconstitution of HOIP in HOIPKO^pool THP1 cells (Fig. 2E,F). Further, LC3B lipidation and STING degradation were unaffected in HOIPKO^pool THP1 cells (Figs. 2E–G and EV2), and shHOIP iBMDMs (Fig. 2H,I) following cGAMP or diABZI activation of STING. These results demonstrate that STING activation induces HOIP synthesis of M1-Ub chains in multiple cell lines, and loss of HOIP does not affect STING activation.

## Loss of STING-mediated VAIL disrupts the perinuclear localization of ubiquitin and HOIP, but not M1-Ub chain formation

While examining the spatial localization of ubiquitin following STING activation, we found that stable overexpression of mEGFP-LC3B in HeLa^STING cells more tightly condenses both ubiquitin and STING into perinuclear foci (Fig. EV3) and increases the detection of M1-Ub ubiquitin chains by immunoblotting following cGAMP treatment (Fig. EV3). As HOIP is not required for STING-induced LC3B lipidation (Figs. 2 and EV2), we questioned whether LC3B lipidation may function upstream of M1-Ub chain formation following STING activation. We previously reported that STING activation-induced LC3B lipidation is distinct from autophagosome formation and occurs through a process we termed V-ATPase-ATG16L1 induced LC3B lipidation (VAIL) at single membrane, Golgi-related vesicles (Fischer et al, 2020). The bacterial effector, SopF, blocks the recruitment of ATG16L1 to the V-ATPase and STING-mediated LC3B lipidation, without affecting autophagy, presenting a useful tool to selectively disrupt VAIL (Xu et al, 2019; Fischer et al, 2020; Xu et al, 2022). Comparison of HeLa^STING cells with or without stable expression of SopF showed no substantial difference in M1-Ub chain formation detected by immunoblotting (Fig. 3A,B), however, ubiquitin foci formation was blocked following cGAMP treatment (Fig. 3C). Similarly, elimination of LC3B lipidation by loss of ATG16L1 caused no effect on STING-mediated M1-Ub chain formation (Fig. 3D,E), whereas ubiquitin and mEGFP-HOIP foci formation were blocked (Fig. 3F,G) in ATG16L1KO HeLa^STING

cells. Further examination of ubiquitin and mEGFP-HOIP localization with saponin extraction in WT and ATG16L1KO HeLa^STING cells following diABZI activation of STING showed that ubiquitin and mEGFP-HOIP appear in small, dispersed punctae, rather than larger, clustered foci (Fig. 3F,G). It has recently been reported that STING may form a proton channel in Golgi membranes upon its activation and trafficking that neutralizes the pH of the Golgi and stimulates LC3B lipidation (Liu et al, 2023; Xun et al, 2023). C53 is a small molecule agonist of STING that is reported to block LC3B lipidation and the neutralization of Golgi pH following STING activation by binding to the putative proton channel region in STING's transmembrane domains (Liu et al, 2023; Xun et al, 2023; Pryde et al, 2021). Treatment of HeLa^STING cells with C53 blocked STING activation-induced LC3B lipidation, and partially reduced M1-Ub chain formation (Fig. EV3) following diABZI treatment. Live imaging of mScarletI-LC3B and the K63-Ub sensor, Vx3-mEGFP, also demonstrated that C53 blocks both LC3B and Vx3 foci formation in the perinuclear region (Fig. EV3). Collectively, these results indicate that the lipidation of LC3B is not required for STING-induced HOIP synthesis of M1-Ub chains but affects the spatial localization of ubiquitin and HOIP. The Golgi pH neutralizing activity of STING may also be involved upstream of both M1-Ub chain formation, ubiquitin localization, and LC3B lipidation, indicating a possible shared signal that induces these processes at the same membranes.

## M1 ubiquitin chains stimulate STING-mediated NFκB signaling

M1 ubiquitin chains are most widely understood to function in immune signaling, particularly through the NFκB pathway (Tokunaga et al, 2009; Rahighi et al, 2009; Haas et al, 2009; Sasaki and Iwai, 2015). Since activation of STING induces NFκB signaling in addition to IRF3-mediated signaling, we asked whether M1 ubiquitin chain formation induced by STING activation mediates NFκB signaling. HeLa cells overexpressing STING displayed differences in IRF3- and NFκB-related gene expression between HOIPKO and OTULIN overexpressing cell lines (Appendix Fig. S1). To avoid potential clonal variation in our HOIPKO line, complications from STING overexpression, and to study a more

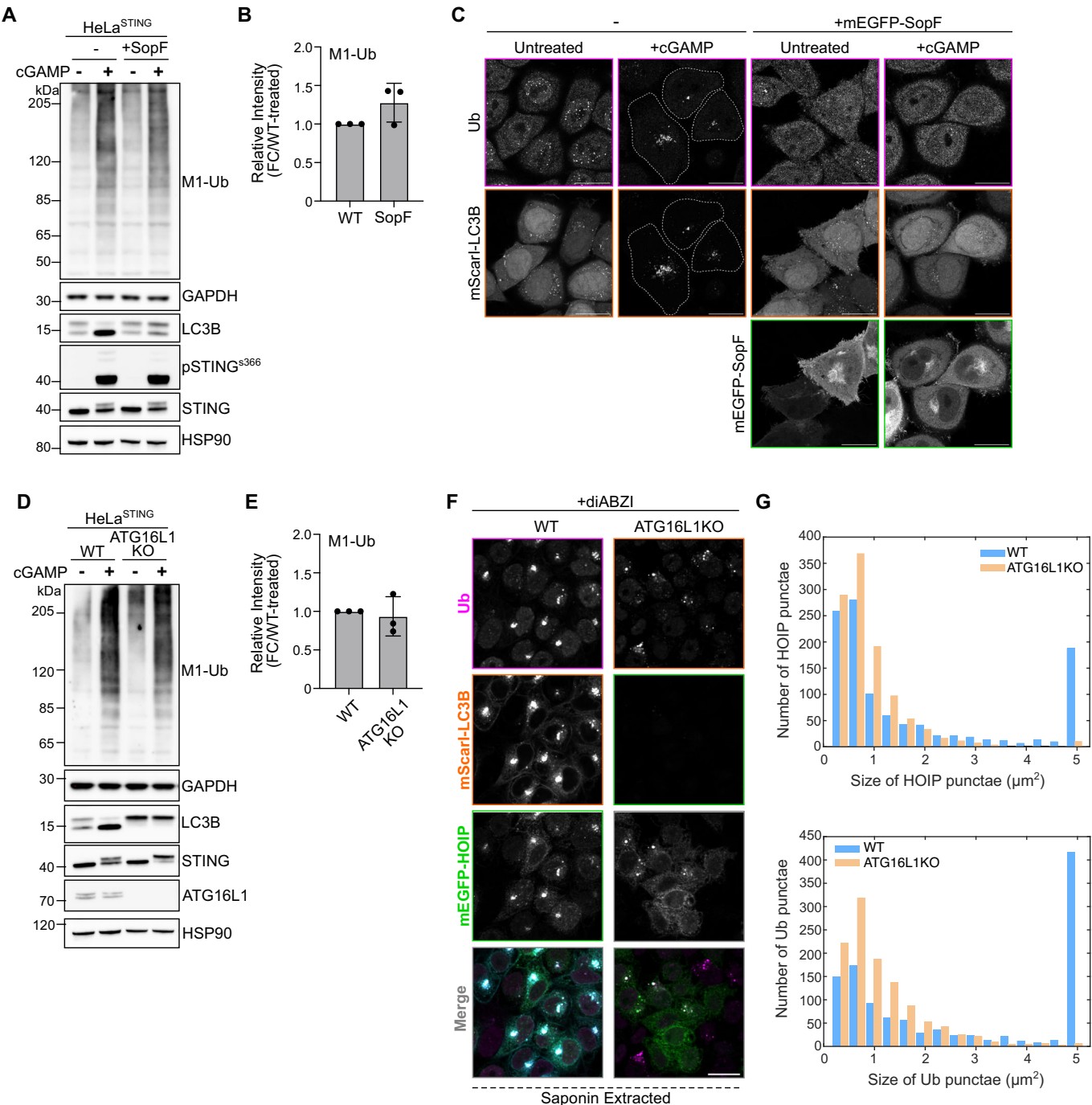

**Figure 3. Loss of STING-mediated VAIL disrupts the perinuclear localization of ubiquitin and HOIP, but not M1-Ub chain formation.**

(A) Representative immunoblots of indicated proteins detected lysates from HeLa[STING] WT and HeLa[STING] with stable overexpression of mEGFP-SopF prepared after treatment with 120 μg/mL cGAMP for 8 h. Immunoblotting was replicated in 3 independent experiments. (B) Quantification of M1-Ub in (A). Values represent the relative intensity of lanes to WT cGAMP-treated lanes. Error bars represent the mean ± s.d. of three independent experiments. (C) Representative Airyscan-processed confocal images of WT HeLa[STING] cells stably expressing mScarletI-LC3B alone or with stable expression of mEGFP-SopF treated with 120 μg/mL of cGAMP for 8 h prior to PFA-fixation and immunostaining with antibodies raised against mono- and poly-ubiquitin chains (Ub). Scale bar = 20 μm. Imaging was replicated in three independent experiments. (D) Representative immunoblots of indicated proteins detected lysates from HeLa[STING] WT and ATG16L1KO cells prepared after treatment with 120 μg/mL cGAMP for 8 h. Immunoblotting was replicated in three independent experiments. (E) Quantification of M1-Ub in (D). Values represent the relative intensity of lanes to WT cGAMP-treated lanes. Error bars represent the mean ± s.d. of three independent experiments. (F) Representative spinning disk confocal images of HeLa[STING]; mEGFP-HOIP (green); mScarletI-LC3B (orange) cells treated with 120 μg/mL of cGAMP for 8 h prior to prior to saponin extraction, PFA-fixation, and immunostaining for ubiquitin (Ub; magenta). Scale bar = 20 μm. (G) Quantification and distribution of the number and size of HOIP (top) and Ubiquitin punctae (bottom) in the experiment represented in (F). Large "punctae" can be interpreted as "foci". Imaging was replicated in three independent experiments. Source data are available online for this figure.

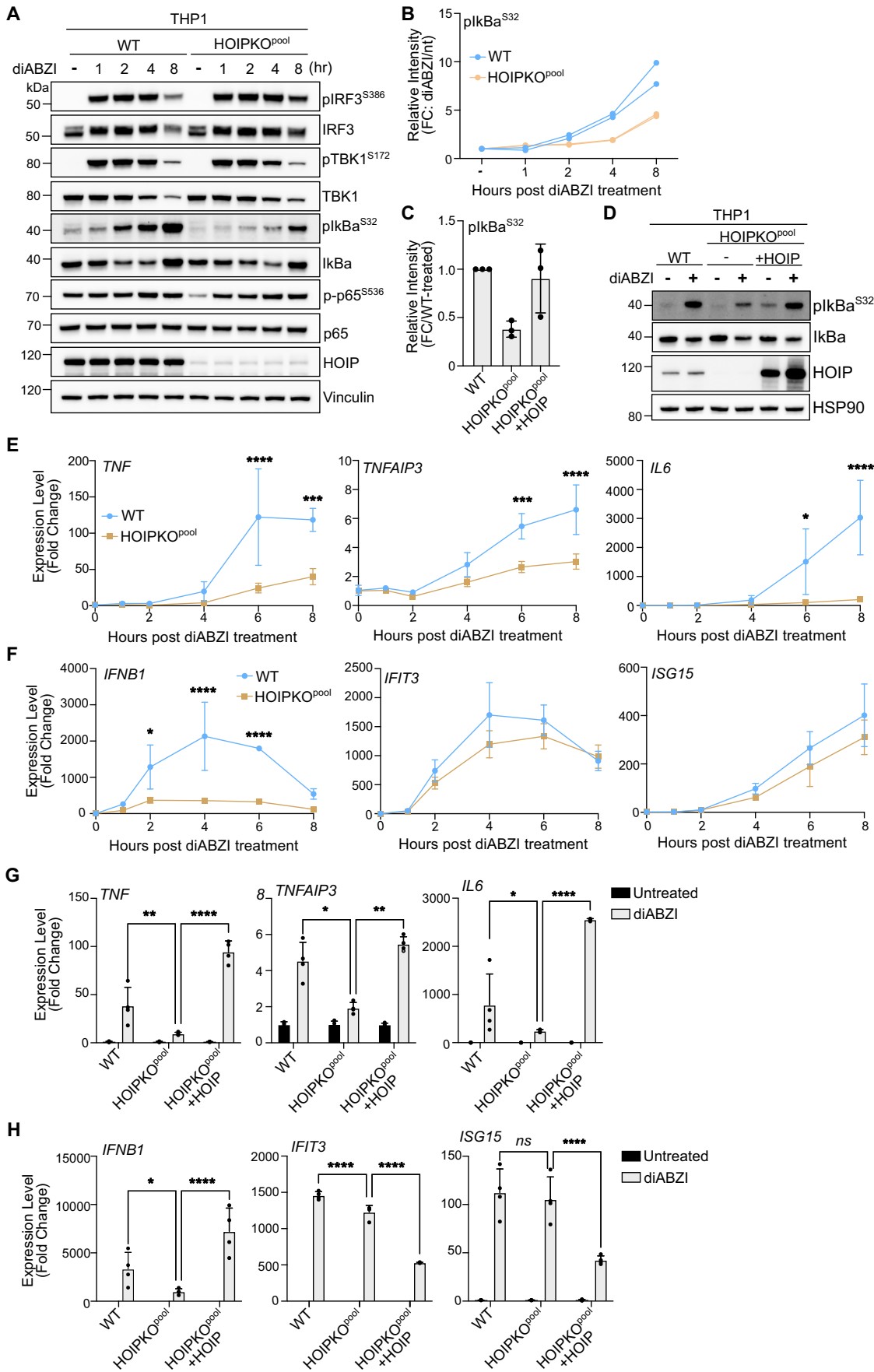

**Figure 4. M1 ubiquitin chains stimulate STING-mediated NFκB-related immune signaling.**

(A) Representative immunoblots of indicated proteins detected in THP1 cell lysates from WT and HOIPKO$^{pool}$ cells prepared following treatment with 1 μM diABZI for 1, 2, 4, and 8 h. Immunoblotting was replicated in three independent experiments. (B) Quantification of pIκBα$^{S32}$ in (A). Values represent the relative intensity of corresponding bands to untreated bands per cell line. Replicates for each condition from two independent experiments are shown. (C) Quantification of pIκBα$^{S32}$ in (D). Values represent the relative intensity of bands to WT-treated bands. Error bars represent the mean ± s.d. of three independent experiments. (D) Representative immunoblots of indicated proteins detected in THP1 cell lysates from WT, HOIPKO$^{pool}$, and HOIPKO$^{pool}$ stably expressing untagged HOIP prepared following treatment with 1 μM diABZI for 4 h. Immunoblotting was replicated in three independent experiments. (E, F) Relative expression changes of indicated NFκB- (E) and IRF3/interferon-related (F) genes detected by quantitative RT-PCR in WT and HOIPKO$^{pool}$ THP1 cells treated with 1 μM diABZI for 1, 2, 4, and 8 h. Quantification of relative expression is from three independent experiments analyzed at the same time. A two-way ANOVA with Sidak's multiple comparisons test was performed on $2^{-\Delta\Delta Ct}$ values. Mean ± s.d. $n = 3$ * < 0.05, ***<0.001, ****<0.0001 (*TNF* 6 h $p$ = <0.0001, 8 h $p$ = 0.0006; *TNFAIP3* 6 h $p$ = 0.0001, 8 h $p$ = <0.0001; *IL6* 6 h $p$ = 0.0117, 8 h $p$ = <0.0001; *IFNB1* 2 h $p$ = 0.0135, 4 h $p$ = <0.0001, 6 h $p$ = <0.0001). (G, H) Relative expression changes of indicated NFκB- (G) and IRF3/interferon-related (H) genes detected by quantitative RT-PCR in THP1 WT, HOIPKO$^{pool}$, and HOIPKO$^{pool}$ stably expressing untagged HOIP cells treated with 1 μM diABZI for 4 h. Quantification of relative expression is from four independent experiments analyzed at the same time. A two-way ANOVA with Tukey's multiple comparisons test was performed on $2^{-\Delta\Delta Ct}$ values. Mean ± s.d. $n = 4$ *<0.05, **<0.01, ****<0.0001 (*TNF* WTvHOIPKO$^{pool}$ $p$ = 0.0019, HOIPKO$^{pool}$vHOIPKO$^{pool}$ + HOIP $p$ = <0.0001; *TNFAIP3* WTvHOIPKO$^{pool}$ $p$ = 0.0333, HOIPKO$^{pool}$vHOIPKO$^{pool}$ + HOIP $p$ = 0.0014; *IL6* WTvHOIPKO$^{pool}$ $p$ = 0.0349, HOIPKO$^{pool}$vHOIPKO$^{pool}$ + HOIP $p$ = <0.0001; *IFNB1* WTvHOIPKO$^{pool}$ $p$ = 0.05, HOIPKO$^{pool}$vHOIPKO$^{pool}$ + HOIP $p$ = <0.0001; *IFIT3* WTvHOIPKO$^{pool}$ $p$ = <0.0001, HOIPKO$^{pool}$vHOIPKO$^{pool}$ + HOIP $p$ = <0.0001; *ISG15* HOIPKO$^{pool}$vHOIPKO$^{pool}$ + HOIP $p$ = <0.0001). Source data are available online for this figure.

immune-relevant cell model, we assessed immune signaling in our HOIPKO$^{pool}$ THP1 monocyte cells. In WT and HOIPKO$^{pool}$ THP1 cells, we assessed markers for both IRF3- and NFκB-related signaling by immunoblotting and gene expression following STING activation by either cGAMP (Fig. EV4) or diABZI (Fig. 4). In WT THP1 cells, robust phosphorylation of IRF3 at serine 386 and TBK1 at serine 172 were detected at the one-hour timepoint post-diABZI treatment (Fig. 4A), and the two-hour timepoint post cGAMP treatment (Fig. EV4). No differences in these events were observed in HOIPKO$^{pool}$ THP1 cells with either treatment (Figs. 4A and EV4). Detection of the degradation of the NFκB inhibitor, IκBα, at the 4-h timepoint post-diABZI (Fig. 4A) or cGAMP (Fig. EV4) treatment in WT cells showed no substantial defect in the HOIPKO$^{pool}$ THP1 cells. However, the phosphorylation of IκBα at serine 32 appeared to be delayed in the HOIPKO$^{pool}$ THP1 cells compared to WT with both treatments (Figs. 4A,B and EV4). Reconstitution of HOIP in HOIPKO$^{pool}$ THP1 cells, rescued the phosphorylation of IκBα following diABZI activation of STING (Fig. 4C,D).

Consistent with reduced upstream activation of the NFκB pathway, transcription of the NFκB-related genes, *TNF*, *TNFAIP3*, and *IL6* were reduced in HOIPKO$^{pool}$ THP1 cells following both diABZI (Fig. 4E) and cGAMP treatments (Fig. EV4). Despite no changes in IRF3 phosphorylation (Fig. 4A), STING activation-induced expression of the IRF3-mediated gene, *IFNB1,* by both diABZI (Fig. 4F) and cGAMP (Fig. EV4) was reduced in HOIPKO$^{pool}$ THP1 cells. However, interferonβ-mediated gene expression was unaffected by loss of HOIP (Figs. 4F and EV4). These results may reflect cross-regulation of *IFNB1* gene induction by NFκB, in line with previous reports (Abe and Barber, 2014). Reconstitution of HOIP in HOIPKO$^{pool}$ THP1 cells, rescued NFκB and IRF3-related gene expression following diABZI treatment (Fig. 4G,H). Consistent with results in THP1 cells, reduced expression of NFκB- and IRF3- genes was also observed with the knockdown of HOIP in mouse immortalized bone marrow-derived macrophages (iBMDMs) following diABZI activation of STING (Fig. EV4). These data indicate that HOIP is important for STING activation-induced NFκB and IRF3-related signaling in human THP1 monocytes, and mouse bone marrow-derived macrophages.

## LC3B lipidation is not required for STING-mediated innate immune responses

Since our discovery that STING activation-induced LC3B lipidation is distinct from autophagy (Fischer et al, 2020), it has been unclear what the function of this process is in STING-mediated innate immunity. Although the loss of LC3B lipidation induced by STING activation did not affect M1-Ub chain formation (Fig. 3), we questioned whether the potential regulation of ubiquitin and HOIP spatial localization we observed (Fig. 3) may impact downstream IRF3- and NFκB-related signaling. To address this question, we examined STING activation in ATG16L1KO THP1 cells. Compared to WT controls, M1-Ub chain formation induced by diABZI treatment was unaffected in ATG16L1KO THP1 cells (Fig. 5A), similar to HeLa cells (Fig. 3C,D). While a defect in STING degradation was detected (Fig. 5A), similarly to previous reports (Gentili et al, 2023), phosphorylation of IRF3, TBK1, and IκBα were unchanged by the loss of ATG16L1 (Fig. EV5). Consistently, no differences between WT and ATG16L1KO THP1 cells were observed in IRF3/interferon- or NFκB-related gene expression following STING activation by diABZI (Fig. 5B,C), despite small, but significant increases in ATG16L1KO cells at the 8 h timepoint in *TNF* and *TNFAIP3* induction (Fig. 5B). We also examined the effect of C53-mediated inhibition of LC3B lipidation on immune-related signaling. Although no substantial differences were observed in diABZI-induced phosphorylation of IκBα with C53 treatment (Fig. EV5), C53 treatment reduced NFκB and IRF3-related gene expression (Fig. EV5). These results indicate that LC3B lipidation induced by STING activation does not regulate M1-Ub chain formation or IRF3- and NFκB-related gene expression. However, the upstream consequences of the reported proton channel activity of STING may regulate both, as well as LC3B lipidation, at Golgi membranes.

## Discussion

Innate immune responses to cytosolic DNA, from viruses, bacteria, mitochondria, or the nucleus are mediated in part by cGAS activation of STING (Ablasser and Chen, 2019). Following the detection of dsDNA, mammalian cGAS synthesizes the cyclic dinucleotide, 2′3′ cGAMP, that binds to STING and induces its

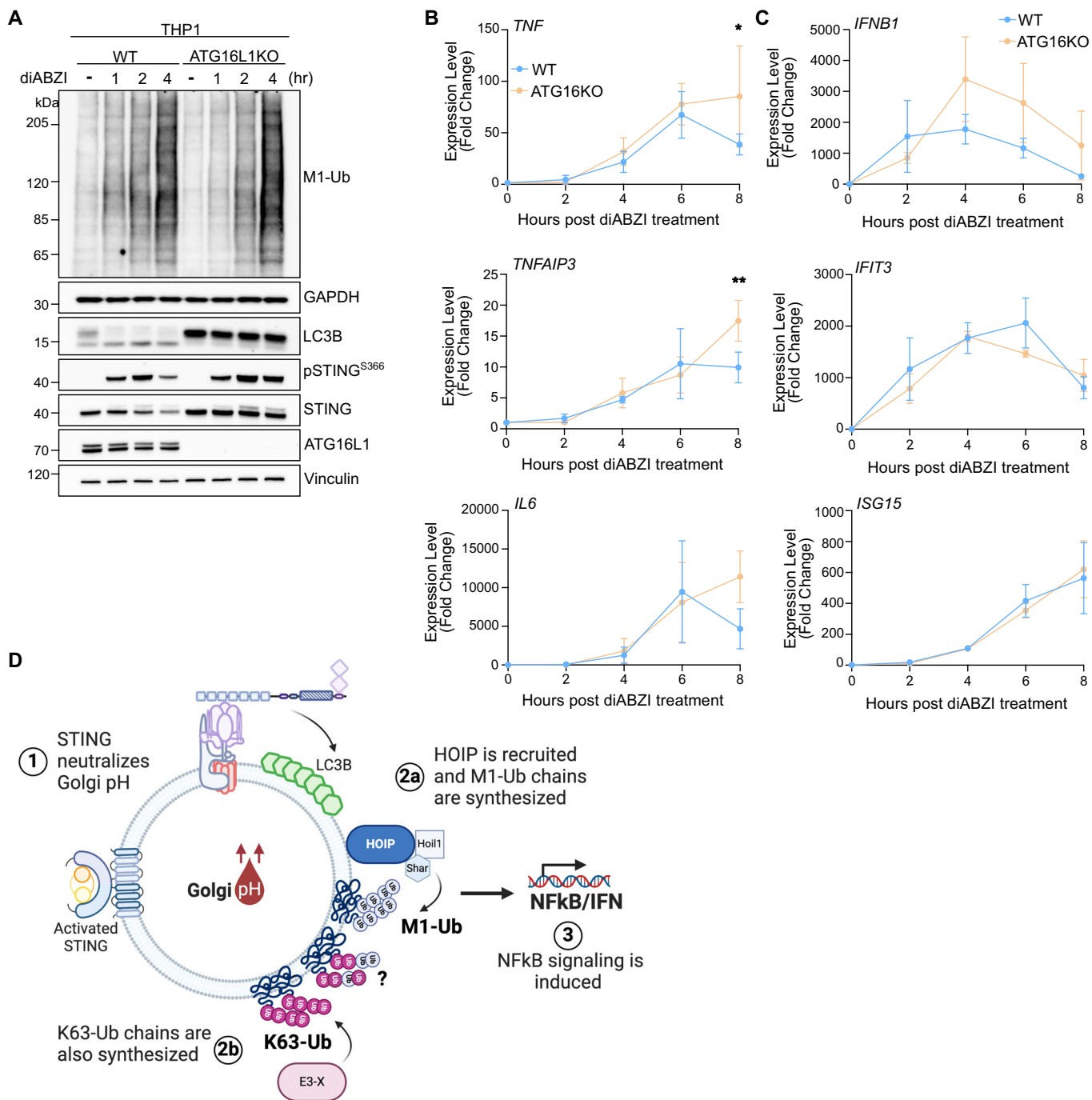

**Figure 5. LC3B lipidation is not required for STING-mediated innate immune responses.**

(A) Representative immunoblots of indicated proteins detected in THP1 cell lysates from WT and ATG16L1KO cells prepared following treatment with 1 μM diABZI for 1, 2, and 4 h. Immunoblotting was replicated in three independent experiments. (B, C) Relative expression changes of indicated NFκB- (B) and IRF3/interferon-related (C) genes detected by quantitative RT-PCR in WT and ATG16L1KO THP1 cells treated with 1 μM diABZI for 2, 4, 6, and 8 h. Quantification of relative expression is from three independent experiments analyzed at the same time. A two-way ANOVA with Sidak's multiple comparisons test was performed on $2^{-\Delta\Delta Ct}$ values. Mean ± s.d. $n = 3$ *<0.05, **<0.01 (*TNF* 8 h $p = 0.0386$; *TNFAIP3* 8 h $p = 0.0081$). (D) Model for STING-induced ubiquitin chain formation at LC3B-associated Golgi membranes and NFκB signaling. Created in BioRender. Fischer, T. (2024). Source data are available online for this figure.

membrane trafficking from the ER to the Golgi apparatus. STING localization at the Golgi and Golgi-related vesicles, initiates IRF3- and interferon-related immune signaling and LC3B lipidation at Golgi membranes, prior to its degradation in the lysosome. STING

activation also induces NFκB transcriptional responses, which confers an important interferon-independent antiviral signature of STING signaling (Ishikawa and Barber, 2008; Abe and Barber, 2014; de Oliveira Mann et al, 2019; Balka et al, 2020; Yum et al,

2021). However, the mechanism of NFκB activation by STING, and the relation to other signaling cascades induced by STING activation and trafficking have remained unclear. We find that STING activation induces the localization of the Linear Ubiquitin Chain Assembly Complex (LUBAC) E3 ligase, HOIP, at LC3B-associated Golgi membranes and its synthesis of M1-linked/linear ubiquitin (M1-Ub) chains to stimulate NFκB-related immune signaling (Fig. 5D). Loss of HOIP prevents M1-Ub chain formation following STING activation by multiple agonists, which impairs IκBα phosphorylation, and the expression of NFκB-related genes. IRF3-related gene expression is also reduced with the loss of HOIP, even though STING, IRF3, and TBK1 phosphorylation remain intact, aligning with potential cross-regulation between IRF3 and NFκB-related gene expression (Abe and Barber, 2014). While our work has revealed a key mechanistic step in NFκB signaling mediated by STING, several questions remain regarding how M1-Ub chains regulate NFκB signaling. The markedly slower kinetics and persistence of M1-Ub chain formation and p-IκBα we report with both cGAMP and diABZI stimulation of STING suggests distinct mechanisms compared to the quickly activated and resolved responses following TNFR or IL1R stimulation (Haas et al, 2009; Emmerich et al, 2013; Tarantino et al, 2014; Ruland, 2011). The recruitment of HOIP to the perinuclear region of cells also raises intriguing questions related to the spatial regulation of LUBAC activity and substrate specificity. Although we were unable to determine whether M1-Ub chains, specifically, are localized in perinuclear foci with LC3B, the localization of HOIP is highly suggestive that M1-Ub may be a ubiquitin species detected at these sites. Overcoming the limitations of linkage-specific ubiquitin cross reactivity (Matsumoto et al, 2012) and more reliable reporters will be necessary to further study the spatial localization of M1-Ub chains following STING activation and in other pathways. K63-Ub chains are one ubiquitin species detectably localized with LC3B foci in the perinuclear region upon STING activation. It will be interesting to determine if these K63-ubiquitin chains are homotypic or heterotypic with M1 ubiquitin chains (i.e., hybrid K63/M1-Ub chains) (Emmerich et al, 2013) or conjugated to specific substrates, and whether the K63-Ub chains play distinct or related roles with M1-Ub chains in STING-induced innate immune signaling. Indeed, STING itself is reported to be conjugated to K63-ubiquitin chains (Tsuchida et al, 2010; Zhang et al, 2012; Ni et al, 2017; Kuchitsu et al, 2023; Prabakaran et al, 2018; Takahashi et al, 2021), which is important for its degradation (Kuchitsu et al, 2023). Unraveling the specific characteristics of these ubiquitin chains will be necessary to understand how they regulate the cGAS-STING pathway.

While our study began by investigating the ubiquitin species localized with LC3B in perinuclear foci we found that STING activation stimulates M1-Ub chain formation and LC3B lipidation in parallel. LC3B lipidation induced by STING is not required for the formation of M1-Ub chains, and HOIP synthesis of M1-Ub chains is not required for LC3B lipidation, in multiple cell lines. However, we observed that loss of LC3B lipidation affects the spatial localization of HOIP and ubiquitin in perinuclear foci in HeLa cells, suggesting that LC3B lipidation may be involved in the remodeling of their resident membranes. Importantly, while LC3B lipidation has been a recognized consequence of STING activation, the function of this process, independently of autophagy, has not been resolved. Here, we show that LC3B lipidation is not required

for STING-mediated IRF3- or NFκB-related immune signaling. These results suggest that the effect of LC3B lipidation on the spatial localization of ubiquitin and HOIP may be dispensable for immune-related transcriptional regulation following STING activation. Indeed, transcription factor activation independent of spatial location would be consistent with previous reports on STING signaling and other pattern recognition receptor pathways (Dobbs et al, 2015; Tan and Kagan, 2019; Landau et al, 2024). Further, we show that inhibition of the recently reported channel activity of STING by the small molecule, C53, that inhibits LC3B lipidation, also partially inhibits STING-mediated M1-Ub and K63-Ub localization, as well as NFκB- and IRF3-related gene expression. These results suggest that the effect of STING on neutralizing Golgi pH may be a shared upstream signal stimulating ubiquitin chain formation and LC3B lipidation at Golgi membranes. Much more work is necessary to unravel the signaling and membrane dynamics between STING, ubiquitin, and LC3B lipidation, at the Golgi and how the combination of these processes shapes STING-mediated innate immunity.

# Methods

**Reagents and tools table**

| Reagent/resource | Reference or source | Identifier or catalog number |
|---|---|---|
| **Experimental models** | | |
| HeLa | ATCC | CVCL_0300 |
| HEK293T | ATCC | #CRL-3216 |
| HOIPKO HeLa | This paper | |
| ATG16L1KO HeLa | Fischer et al, 2020 | |
| WT THP1 (match to HOIPKO) | Synthego-ATCC | |
| HOIPKO THP1 | This paper (Synthego) | |
| ATG16L1KO THP1 | This paper (Synthego) | |
| **Recombinant DNA** | | |
| pSpCas9(BB)-2A-Puro (PX459) V2.0 | Addgene | #62988 |
| pLKO.1 | Addgene | #8453 |
| pTRIP_BFP-P2A-STING | Addgene | #102597 |
| pHAGEb_mEGFP-LC3B | This paper | |
| pHAGEb_mEGFP-HOIP | This paper | |
| pHAGEb_HOIP_IRES_mCh-3xNLS | This paper | |
| pHAGEb_mScar-LC3B | This paper | |
| pHAGEb_mEGFP-OTULIN | This paper | |
| pHAGEb_mEGFP-SopF | This paper and Fischer et al, 2020 | |
| **Antibodies** | | |
| Ms Mono- and polyubiquitinylated conjugates (UBCJ2) | Enzo | ENZ-ABS840 |
| Ms Anti-STING (OTI4HI) | Thermo Scientific | MA5-26030 |
| Rb Anti-STING/TMEM173 | Proteintech | 19851-1-AP |

 

| Reagent/resource | Reference or source | Identifier or catalog number |
|---|---|---|
| Anti-Ubiquitin, Clone VU-1 | Life Sensors | VU-0101 |
| Rb Anti-Ubiquitin, K48-Specific, Clone Apu2 | Millipore Sigma | ZRB2150 |
| Rb Anti-Ubiquitin, Lys63-specific, Clone Apu3 | Millipore Sigma | 05-1308 |
| Rb Anti-Linear Ubiquitin, clone 1E3 ZooMab | Sigma-Aldrich | ZRB2114 |
| Rb Anti-GAPDH | Sigma-Aldrich | G9545 |
| Rb Anti-HOIP/RNF31, E6M5B | CST | 99633S |
| Rb Anti-RNF31 | Fortis Life Sciences | A303-560A-T |
| Rb Anti-Ranbp-Type and C3Hc4-Type Zinc Finger Containing 1 (HOIL-1) | Proteintech | 26367-1-AP |
| Rb Anti-SHARPIN | Proteintech | 14626-1-AP |
| Rb Anti-LC3B | Sigma-Aldrich | L7543 |
| Rb Anti-phopsho-STING(S366), D7C3S | CST | 19781 s |
| Rb Anti-STING(D2P2F) | CST | 13647 |
| Rb Anti-HSP90(a/b), H-114 | SCBT | sc-7947 |
| Rb Anti-TNFAIP3-Interacting Protein 1 (TNIP1) | Proteintech | 15104-1-AP |
| Rb Anti-OPTN | Proteintech | 10837-I-AP |
| Rb Anti-IkBa, 44D4 | CST | 4812S |
| Rb Anti-phospho-IkBa(S32), 14D4 | CST | 2859S |
| Rb Anti-NFkB p65 (D14E12) XP | CST | 8242 |
| Rb Anti-phosho-NFkB p65 (S536) (93H1) | CST | 3033 |
| Rb Anti-TBK1/NAK | CST | 3013 |
| Rb phospho-TBK1/NAK(S172) (D52C2) XP | CST | 5483 |
| Rb Anti-IRF3 (D83B9) | CST | 4302 |
| pIRF3(S386) | Abcam | ab76493 |
| Rb Anti-Vinculin (E1E9V) XP | CST | 13901S |
| Ms Anti-Actin, Clone C4 | Sigma-Aldrich | MAB1501 |
| Goat Anti-Rabbit IgG (H + L) Cross-Adsorbed, Alexa Fluor 647 | Invitrogen | A-21244 |
| Goat Anti-Mouse IgG (H + L) Cross-Adsorbed, Alexa Fluor 647 | Invitrogen | A-21235 |
| Goat Anti-Rabbit IgG (H + L) Cross-Adsorbed, Alexa Fluor 546 | Invitrogen | A-11035 |
| Goat Anti-Mouse IgG (H + L) Cross-Adsorbed, Alexa Fluor 546 | Invitrogen | A-11003 |
| Rabbit IgG, HRP-Linked F(ab')2 Fragment, Donkey | Cytiva | NA9340 |
| Mouse IgG, HRP-Linked F(ab')2 Fragment, Sheep | Cytiva | NA9310 |
| **Oligonucleotides and other sequence-based reagents** | | |
| qPCR primers sequences | This paper | Appendix Table 1 |
| CRISPR gRNA sequences | This paper | Appendix Table 2 |

| Reagent/resource | Reference or source | Identifier or catalog number |
|---|---|---|
| shRNA Target sequences | This paper | Appendix Table 3 |
| **Chemicals, Enzymes and other reagents** | | |
| 2'3' cGAMP Sodium Salt | Chemietek | CT-CGMAP |
| Monensin sodium salt, 90-95%(TLC) | Sigma-Aldrich | M5273 |
| TNFa | Peprotech | 300-01 A |
| diABZI STING agonist-1 (trihydrochloride) | MedChem Express | HY-112921B |
| diABZI STING agonist (Compound 3) | Selleck Chemicals | S8796 |
| STING agonist C53 | Tocris | 7741 |
| Puromycin | Invivogen | ant-pr-1 |
| Saponin | Sigma-Aldrich | 47036 |
| Shield1 | TakaraBio | 632189 |
| Doxycycline | Sigma-Aldrich | D3447 |
| Quick-RNA MiniPrep Plus kit | Zymo Research | R1058 |
| High-Capacity cDNA Reverse Transcription Kit | Thermo Scientific | 4368814 |
| PowerUp SYBR Green Master Mix | Thermo Scientific | A25742 |
| Control Magnetic Beads (negative control) | Lifesensors | UM-400M-1000 |
| TUBE 1 (High-capacity magnetic beads) | Lifesensors | UM501M |
| Hoechst | Thermo Scientific | H1399 |
| Jetoptimus | Polyplus | 101000006 |
| NEBuilder HiFi DNA Assembly Cloning Kit, with Competent Cells | New England Biolabs | E5520S |
| DMEM | Thermo Fisher | #31053028 |
| Fetal Bovine Serum - Premium Select, Heat inactivated | R&D | S11550H |
| Sodium Pyruvate | Thermo Fisher | #11360070 |
| GlutaMAX | Thermo Fisher | #35050061 |
| RPMI (ATCC modification) | ATCC | 30-2001 |
| OptiMEM | Gibco | 31985062 |
| **Software** | | |
| Zen | Zeiss | |
| ImageLab | Bio-Rad | |
| FIJI | Schindelin et al, 2012 | |
| Prism | GraphPad | |
| **Other** | | |
| 8 well-chambered coverglass with #1.5 high-performance coverglass | Cellvis | C8-1.5H-N |
| 96-well glass bottom plates | DOT Scientific Inc | MGB096-12-LG-L |
| Surepage 4–12% Bis-Tris gels, 10-, 12-, 15-well | Genscript | |
| Nupage 4–12% Bis-Tris gels, 20 well | Thermo Scientific | WG1402 |

| Reagent/resource | Reference or source | Identifier or catalog number |
|---|---|---|
| Nupage 3–8% Tris-Acetate gels, 1.0 mm, 15-well | Thermo Scientific | EA03755 |
| S-Trap Columns | Protifi | C02-micro |

Detailed information for cell lines, plasmids, antibodies, chemicals, enzymes, and other reagents/tools are provided in the Reagents and Tools Table. Primer sequences and CRISPR and shRNA sequences are provided in Appendix Tables 1–3.

## Methods and protocols

### Cell culture

HeLa (ATCC), HeLa FRT/TREX, HEK293T (ATCC), and immortalized mouse bone marrow-derived macrophage (iBMDM) cells were cultured in Dulbecco's Modified Eagle Medium (DMEM; Thermo Fisher) supplemented with 10% (v/v) fetal bovine serum (FBS; R&D), 1 mM sodium pyruvate (Thermo Fisher), and 2 mM GlutaMAX (Thermo Fisher). THP1 (Synthego-ATCC) cells were cultured in Roswell Park Memorial Institute (RPMI) 1640 (ATCC). HeLa FRT/TREX cells stably expressing an inducible version of the Vx3-EGFP sensor previously reported in (Sims et al, 2012) with a DD degron fused to the N-terminus (pcDNA5-FRT/TO-DD-Vx3-EGFP) were kindly provided by Tingting Yao and Robert Cohen. iBMDMs were kindly shared by Howard Young and previously reported in (Blasi et al, 1987, 1989).

### Cloning and generation of stable cell lines

pHAGEb vectors were modified from the pHAGE (HIV-1 Gustavo George Enhanced) vector. Plasmids were generated by PCR amplification of open reading frames (ORFs) in cDNA and ligation into a linearized pHAGEb vector using a Gibson assembly kit (New England Biolabs). dH5a or Stable competent E. coli cells (New England Biolabs) were then transformed with assembled plasmids and grown on LB agar plates with appropriate antibiotics. Single colonies were selected, expanded in antibiotic containing liquid culture, and screened for successful insertion of the amplicon by diagnostic restriction enzyme digest. Selected plasmids were further validated by Sanger sequencing. Complete plasmid sequences are available upon request.

HeLa cell lines with stable overexpression of indicated genes were generated by lentivirus infection. Lentivirus was generated using HEK293T cells and lentiviral system plasmids (pHDM-G, pHDM-HGPM2, pHDM-tat1B, and pRC-CMV-rev1B) using a protocol detailed in Wang, 2020 (Wang, 2020). HEK293T cells were transfected with the lentiviral system plasmids (500 ng each), plasmid DNA (1 ug), and PEI prepared in OptiMEM media (Gibco). Following an initial exchange for fresh media 16–24 h after transfection, lentiviral conditioned media was harvested 28–72 h post-transfection and filtered through a 0.45 um PVDF syringe filter prior to transduction of target cells. Target cells were transduced by incubating cells with generated lentivirus and 8 ug/mL Polybrene for 24 h. At least 3 days following transduction, generated stable cell lines were sorted based on the expressed fluorescent proteins using Fluorescence-Activated Cell Sorting (FACS) to obtain homogenous cell populations at empirically determined optimal protein expression.

### Generation of CRISPR knockout cell lines

For HOIPKO HeLa cells, gRNAs were annealed and ligated into a BbsI-digested pSpCas9(BB)-2A-Puro (PX459) V2.0 vector (Addgene #62988). Assembled plasmid DNA was expanded as described above. HeLa cells were transfected with the gRNA-containing plasmid using the JetOptimus (Polyplus) transfection reagent, and subsequently treated with 1 ug/ml puromycin (InvivoGen) for 2 days to select for cells expressing the plasmid. Isolation of single cells from the puromycin-resistant pool was obtained by limiting dilution and clonal expansion in 96-well plates. Individual clones were then screened for knockout edits by PCR amplification of the target locus from extracted genomic DNA and Sanger sequencing of the amplicon. Sequence data were analyzed using the Inference of CRISPR Edits (ICE) tool from Synthego to identify clones with frame-shifting insertions or deletions (InDels). Selected clones were finally validated by measuring protein abundance detected by immunoblotting. ATG16L1KO HeLa cells are from Fischer et al, 2020. HOIPKO and ATG16L1KO THP1 and Mock transfected WT cells were purchased as an express pooled cell line from Synthego. Details for gRNA sequences for each KO cell line are in Appendix Table 2.

### shRNA Knockdown cell lines

For shHOIP cells, the target shRNA sequence for mouse RNF31 (Appendix Table 3) was adapted, annealed, and ligated into EcoRI and AgeI-digested pLKO.1 vector as per the Addgene protocol (addgene: #8453). Assembled plasmid DNA was expanded, and lentivirus was produced as described above for both the shRNA target-containing vector (shHOIP) and an empty vector (shCtrl). iBMDMs were incubated with viral inoculum for 7 h. Seventy-two hours after removal of viral inoculum, cells were treated with 10 ug/mL puromycin for 6 days, and then validated for knockdown at the protein level by immunoblotting.

### Cell treatment with STING agonists

For treatments of different STING agonists (cGAMP, diABZI, and C53), 12–24 h following cell plating for each experiment (more details below), the cell culture medium was replaced, and agonists were added to the fresh medium. No transfection or permeabilization reagents were used.

### Immunofluorescence

Cells were plated onto #1.5 chambered coverglass (Cellvis) or in 96-well glass plates (DOT Scientific Inc) 18–24 h before experiments were performed. Following indicated treatments, cells were fixed with prewarmed 4% paraformaldehyde (PFA; Electron Microscopy Services) for 10–15 min at 37 °C. For general immunofluorescence procedures, fixed cells were rinsed with 1xPBS and permeabilized with 0.5% Triton-X 100 (Sigma) in 1xPBS for 5 min prior to blocking for 1 h in buffer containing 3% goat serum, 1% BSA, and 0.1% Tween-20 in 1xPBS. Cells were then incubated with primary antibody prepared in blocking buffer overnight at 4 °C and then washed with 1xPBS prior to incubation with secondary antibodies in blocking buffer for 1 h at room temperature.

For linkage-specific ubiquitin antibodies, cells were incubated in blocking buffer overnight at 4 °C, and then incubated in primary antibodies prepared in blocking buffer for 1 h at room temperature prior to the washing and secondary antibody incubation steps per the recommended protocol described in Newton et al, 2012

 

(Newton et al, 2012). Following immunostaining, cells were stained with Hoechst dye prior to imaging.

For saponin extraction, cells were rinsed with ice-cold HBSS, permeabilized with Saponin Extraction Buffer (SEB; 80 mM PIPES, pH 6.8, 1 mM MgCl$_2$, 1 mM EGTA, 0.1% Saponin (w/v)) for 2 min, then 4 min on ice, and washed again 2x with HBSS prior to fixation as described above (Tarantino et al, 2014).

### Live-cell imaging

For live cell imaging, cells were plated in 96-well glass plates (DOT Scientific Inc) 18–24 h before experiments were performed. For Vx3-EGFP (K63-Ub Sensor), cells were treated with Doxycycline (1 ug/mL) and the Shield1 ligand (500 nM) at the time of plating and at least 24 h prior to experiments. Image acquisition began immediately after indicated treatments and occurred every 30 min over a 12-h time course. Imaging was performed in a live-cell chamber at 37 °C, 5% CO$_2$, and constant humidity.

### Microscopy

For super-resolution imaging, fixed and/or immunostained cells were imaged on a Zeiss LSM 880 Airyscan Confocal Microscope using a Zeiss 63×1.4 NA Plan-Apochromat objective, equipped with a Piezo high-precision stage at room temperature. Z-stack images were acquired with 405-, 488-, 561-, and 633-nm lasers, MBS 488/561/633 and MBS 405 beam splitters, Zeiss BP 570-620 + LP 645, BP 420-480 + BP 495-550, and BP 420-480 + LP 605 emission filters, and the Airyscan detector in frame scan mode using the Zen software platform (Carl Zeiss Microscopy). Images were then processed for Airyscan 3D deconvolution using Zen software and prepared for publication using FIJI open-source software (Schindelin et al, 2012). The upper limits of the pixel display range were adjusted to improve brightness in representative images.

For quantitative and live cell imaging, fixed and immunostained cells, or live cells, were imaged on a Nikon Ti-2 CSU-W1 spinning disk confocal microscope using a 40 × 1.15 NA water objective, equipped with an automated TI2-N-WID water immersion dispenser. Images were acquired with 405-, 488-, 561-, and 633-nm lasers and 698/70 nm emission filters using the Nikon elements AR microscope imaging software.

### Image analysis

For Pearson Correlation Coefficient analysis, custom MATLAB scripts were used that performed automated background subtraction and cell segmentation by intensity and size. Individual cells were segmented by performing a Gaussian blur on the LC3B channel, then performing a water-shedding algorithm to separate the blurred peaks. The separating lines from watershedding were then imposed on the cell mask to separate connected components to isolate single cells. The Pearson's Correlation Coefficient was calculated for the pixels within each cell, and then the median value was found for each image. The median value was calculated between each image taken for each well of a 96-well plate, and then averages and standard deviations were calculated between wells.

For percent cells with foci, custom MATLAB scripts performed similar processing and cell segmentation to Pearson's Correlation analysis, but a secondary segmentation for punctate structures was performed. To find punctae and foci, cell-sized regions were equalized by dividing the image by a Gaussian blur of the image,

then pixels with high intensity in both the original image and the equalized image were found and filtered by size and, in some cases, circularity. The number of pixels positive in the foci mask were then counted for each connected component in the segmented single-cell mask and, if above a threshold, were considered cells positive for foci.

For percent foci positive for ubiquitin, we used custom MATLAB scripts to identify LC3B foci as above, then treated each as individual connected components in a loop. Each component underwent a morphological dilation, and the original mask was subtracted to generate a "donut" shape with no overlapping foci. Then a ratio was found between the foci component and the corresponding dilation to find if there is a higher intensity in the foci compared to near the foci. Foci with a ratio greater than the threshold of 1.75 were considered positive for ubiquitin or a specific ubiquitin chain for which they were stained. All MATLAB scripts are available upon request. Please see the schematic in Appendix Fig. S2 for MATLAB workflow.

### Immunoblotting

Cells were plated in 12-well or 6-well plates 18–24 h prior to experiments. Following indicated treatments, cells were washed with ice-cold HBSS, lysed in ice-cold 1x LDS sample buffer (Genscript) supplemented with cOmplete Protease Inhibitor cocktail (Roche), and boiled immediately at 99 °C for 10 min. Protein quantitation per sample was obtained using Pierce BCA protein assay kit (Thermo Fisher). Dithiothreitol (DTT; Sigma) was added to each sample at a final concentration of 100 mM just before gel loading.

For general immunoblotting procedures, 20–30 ug of total protein per sample was loaded into wells of 4–12% Bis-Tris gels (Genscript or Invitrogen) and separated in MES or MOPS buffer ran at 65 v for 30 min, then 125 v until the dye front reached the bottom of the gel. Separated proteins were then transferred onto 0.45 um Nitrocellulose membranes in Bio-Rad Transblot Turbo Transfer Buffer with 20% EtOH using the Bio-Rad Trans-Blot Turbo Transfer system (semi-dry). Membranes were blocked in 5% milk 1xPBST at room temperature (RT) for 1 h prior to primary antibody incubation in 3% BSA 1xPBST at 4 °C overnight. Membranes were then thoroughly washed in PBST, incubated in HRP conjugated secondary antibodies raised against the appropriate species in 5% milk PBST for 1 h at room temperature, and finally washed. HRP signal was developed using either Amersham ECL Prime (Cytiva) or SuperSignal West Femto ECL (Thermo Scientific) and detected using a ChemiDoc Imaging System (Bio-Rad). Images were analyzed using ImageLab (Bio-Rad).

For the detection of linkage-specific ubiquitin, a modified version of the protocol described in Newton et al, 2012 (Newton et al, 2012) was used. About 10–15 ug total protein was loaded into wells of 3–8% tris-acetate gels and separated in tris-acetate buffer (Invitrogen), ran at 65 v for 30 min and then 125 v until the dye front reached the bottom of the gel. Separated proteins were then transferred onto 0.45 um a nitrocellulose membrane at 30 v for 2 h in tris-glycine buffer (Towbin formulation) supplemented with 10% MeOH using the XCell SureLock blot module system (semi-wet). Membranes were blocked in 5% milk 1xPBST overnight at 4 °C, then incubated in primary antibodies in 5% milk 1xPBST for 1 h at room temperature. Membranes were then thoroughly washed in PBST, incubated in HRP conjugated secondary antibodies raised

against the appropriate species in 5% milk PBST for 1 h at room temperature, and finally washed. HRP signal was developed using SuperSignal West Femto ECL and detected and analyzed as described above.

For quantification of immunoblots, raw images were analyzed using ImageLab (Bio-Rad) to detect band volume or lane volume (phosho and Ub blots). All band or lane values were normalized to the corresponding internal loading control (GAPDH, Vinculin, or HSP90). Relative values were then calculated as indicated in the figures and plotted using Prism 9.

### LC-MS/MS quantification of ubiquitin linkages

HeLa cells were plated in a 15 cm dish ~48 h prior to experiments. Following treatment, cells were washed 2x with ice cold 1xPBS supplemented with N-ethymaleimide (NEM; 5 mM), scraped from the dish, and centrifuged at $2000 \times g$ for 2 min at 4 °C. The supernatant was removed, and the cell pellets were resuspended in ice-cold lysis buffer (50 mM Tris-HCl, pH 7.5, 150 mM NaCl, 1% NP-40, 1 mM EDTA, 10% Glycerol) supplemented with the deubiquitylase and protease inhibitors (100 uM PR-619, 5 mM O-phelanthroline, 5 mM NEM, and cOmplete protease inhibitor cocktail). For the preparation of peptides from cell lysates, a modified protocol for S-Trap spin column digestion from Protifi was used. Approximately 40 ug of protein from the cell lysates was solubilized in either a mixture of 5% SDS, 8 M Urea, and 100 mM TEAB or 10% SDS and 100 mM TEAB, and mixed at 50 °C for 5 min. After solubilization, samples were reduced with 0.1 M TCEP for 15 min at room temperature, then alkylated with 0.22 M NEM for 15 min at room temperature in the dark. Samples were then acidified with 12–21% aqueous phosphoric acid prior to digestion with Trypsin/LysC (Promega) and then added immediately to an S-Trap column (Protifi) loaded with S-Trap binding buffer (90% MeOH and 100 mM TEAB). Following initial centrifugation at $4000 \times g$ to trap the proteins, the column was washed with S-trap binding buffer, and then either incubated overnight at 37 °C or for 2 h at 47 °C in digestion buffer (Trypsin-LysC and 50 mM TEAB) for complete protein digestion. After digestion, the column was rehydrated with 50 mM TEAB, and digested peptides were eluted with 0.2% aqueous formic acid, and then a mixture of 0.2% aqueous formic acid and 50% acetonitrile. Peptides eluted from the S-Trap were dried under vacuum and stored at −20 °C until analysis. For analysis, each sample was resuspended in 0.1% TFA, a nanodrop reading was taken at UV280 to normalize loading. NanoLC-MS/MS analysis of tryptic peptides was carried out with a Thermo Scientific Fusion Lumos tribrid mass spectrometer interfaced to a UltiMate3000 RSLCnano HPLC system (Thermo Scientific). For each analysis, ~1 µg of the tryptic digest was loaded and desalted in an Acclaim PepMap 100 trapping column (75 µm, 2 cm) at 4 µL/min for 5 min. Peptides were then eluted into a Thermo Scientific Acclaim PepMap™ 100 column, (3 µm, 100 Å, 75 µm × 250 mm) and chromatographically separated using a binary solvent system consisting of A: 0.1% formic acid and B: 0.1% formic acid and 80% acetonitrile at a flow rate of 300 nL/min. A gradient was run from 5% B to 37.5% B over 60 min, followed by a 5-min wash step with 90% B and 10-min equilibration at 5% B before the next sample was injected. The orbitrap Lumos mass spectrometer operated in unscheduled parallel reaction monitor (PRM) mode. Precursor masses were detected in the Orbitrap at $R = 120{,}000$ (m/z 200). The charge state with the strongest signal of each ubiquitylated ubiquitin peptide was added to the target list. Isolation window was 1.2 m/z. MS/MS spectra were acquired in the orbitrap with $R = 30{,}000$

(m/z 200), with an AGC target 500%. Collision energy was 28%. Data were processed using Skyline (MacLean et al, 2010) for quantification. Peak detection and integration were manually validated for each peptide before quantification results were exported to Excel.

### Quantitative real-time PCR

Total RNA from $5–10 \times 10^5$ cells was extracted using Quick-RNA MiniPrep Plus kit (Zymo Research) followed by reverse transcription using High-Capacity cDNA Reverse Transcription Kit (Thermo Fisher Scientific). Equal amounts of cDNA and corresponding primers were used for qPCR using SYBR Green Master Mix (Thermo Fisher Scientific) and a CFX384 real-time system/C100 Touch Thermal Cycler (Bio-Rad). For each biological sample, the Ct value of the gene interested was normalized against the b-actin Ct to calculate ΔCt. Each ΔCt was normalized to the average ΔCt of untreated samples to generate the ΔΔCt value. Relative gene expression was then analyzed using the $2^{-\Delta\Delta Ct}$ formula and plotted in figures (Livak and Schmittgen, 2001). Details for PCR primers can be found in Appendix Table 1.

### Statistics

For quantitative RT-PCR, one-way or two-way ANOVA was performed on the $2^{-\Delta\Delta Ct}$ values with a Tukey's or Sidak's multiple comparisons test. Statistical analyses were performed using Prism (GraphPad).

## Data availability

All cell lines and plasmids are available upon request and materials transfer agreement. MATLAB codes are available upon request. MS data: MassIVE MSV000092940; ProteomeXchange PXD045603.

The source data of this paper are collected in the following database record: biostudies:S-SCDT-10_1038-S44318-024-00291-2.

## Peer review information

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

## Acknowledgements

We would like to thank the NHLBI Flow Cytometry Core, the NINDS Light Microscopy Core, and the NIDCR Mass Spectrometry Facility (ZIA DE00075) for their excellent assistance. We would like the thank Howard Young, NIH Scientist Emeritus at NCI and the immunology community at NIH for sharing resources and insights on NFκB signaling. We would also like to thank the Youle lab members for their feedback and helpful discussions. Funding was provided by the Intramural Research Program (IRP) of the National Institute of Neurological Disorders and Stroke (NINDS; RJY) and the National Institute of Dental and Craniofacial Research (NIDCR; AW).

## Author contributions

**Tara D Fischer**: Conceptualization; Resources; Data curation; Formal analysis; Supervision; Validation; Investigation; Visualization; Methodology; Writing—original draft; Project administration; Writing—review and editing.
**Eric N Bunker**: Data curation; Software; Formal analysis; Validation; Investigation; Visualization; Methodology; Writing—review and editing. **Peng-Peng Zhu**: Data curation; Formal analysis; Validation; Investigation; Visualization; Methodology; Writing—review and editing. **Francesco Scavone**: Resources; Writing—review and editing. **Mahan Hadjian**: Investigation; Writing—review and editing. **Eunice Dominguez-Martin**: Investigation; Writing—review and editing. **Franscesco Scavone**: Resources; Writing—review and editing. **Robert Cohen**: Resources; Supervision; Writing—review and editing. **Tingting Yao**: Resources; Supervision; Writing—review and editing. **Yan Wang**: Data curation; Formal analysis; Supervision; Validation; Investigation; Methodology; Writing—review and editing. **Achim Werner**: Formal analysis; Supervision; Funding acquisition; Methodology; Writing—review and editing. **Richard J Youle**: Conceptualization; Supervision; Funding acquisition; Writing—original draft; Project administration; Writing—review and editing.

Source data underlying figure panels in this paper may have individual authorship assigned. Where available, figure panel/source data authorship is listed in the following database record: biostudies:S-SCDT-10_1038-S44318-024-00291-2.

## Funding

## Disclosure and competing interests statement

The authors declare no competing interests.

# Expanded View Figures

**Figure EV1.   Total-Ub and K63-Ub co-localization at LC3B foci following STING activation.**

(**A**) Representative spinning disk confocal images of HeLa^STING; mEGFP-LC3B (green) cells treated with 120 μg/mL of cGAMP for 8 h prior to PFA-fixation and immunostaining for mono- and poly-ubiquitin chains (Ub; magenta), and STING (cyan). Scale bar = 20 μm. Corresponding to quantification in Fig. 1B. (**B**) Representative spinning disk confocal images of HeLa^STING; mEGFP-LC3B (green) cells treated with 1 μM diABZI for 4 h prior to PFA-fixation and immunostaining for mono- and poly-ubiquitin chains (Ub; magenta), and STING (cyan). Scale bar = 20 μm. Corresponding to quantification in Fig. EV1C. (**C**) Quantification of the percentage (%) of cells positive for mEGFP-LC3B foci and immunostained Ub foci (left panel), and the percentage (%) of mEGFP-LC3B foci with overlapping immunolabeled signal for Ub, STING, or both (right panel) from experiments represented in Fig. EV1B. Error bars represent ±s.d. from three replicates analyzed in the same experiment. Imaging was replicated in three independent experiments. (**D, E**) Representative spinning disk confocal images of HeLa^STING; mEGFP-LC3B cells treated with 120 μg/mL of cGAMP for 8 h prior to PFA-fixation and immunostaining for K48- and K63-ubiquitin chains. Scale bar = 20 μm. (**F**) Quantification of the percentage (%) of mEGFP-LC3B foci with overlapping signal for immunolabeled K48- and K63-ubiquitin chain from experiments represented in Fig. EV1D, E. Mean ± s.d. from $n = 3$ replicates analyzed in the same experiment. Imaging was replicated in three independent experiments. (**G**) Pearson's correlation coefficient of mScarletI-LC3B and Vx3-EGFP over time in the live imaging experiment represented in Movie EV1. FRT/TREX HeLa cells stably expressing FRT/TO-DD-Vx3-EGFP, BFP-P2A-STING, and mScarletI-LC3B were incubated with 1 μg/mL Doxycycline and 500 nM Shield1 for 24 h prior to treatment with either 120 μg/mL cGAMP or 1 μM diABZI and imaging every 30 min for 12 h on a spinning disk confocal microscope. Error bars represent ± s.d. from three replicates analyzed in the same experiment. Imaging was replicated in three independent experiments.

▶

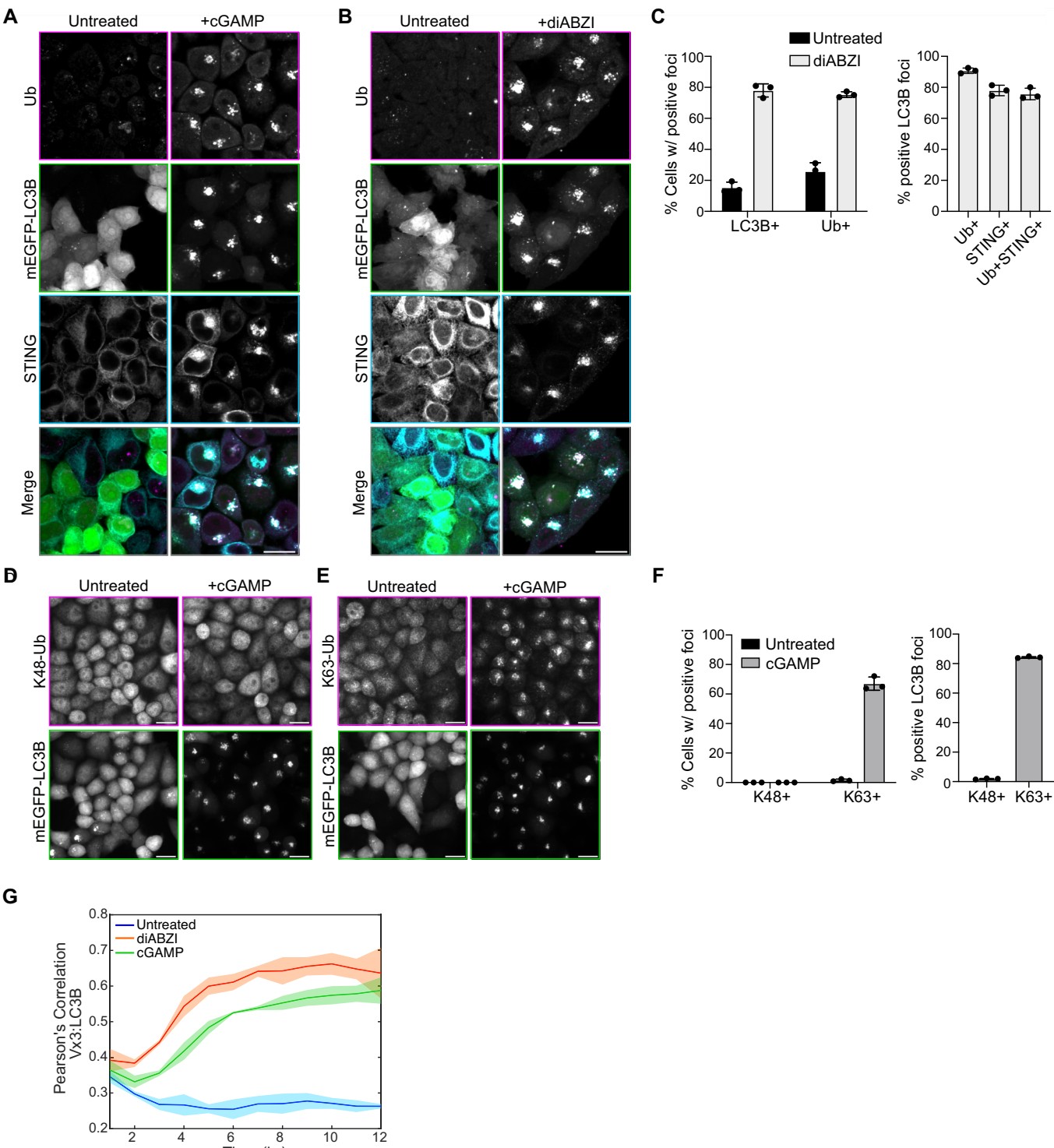

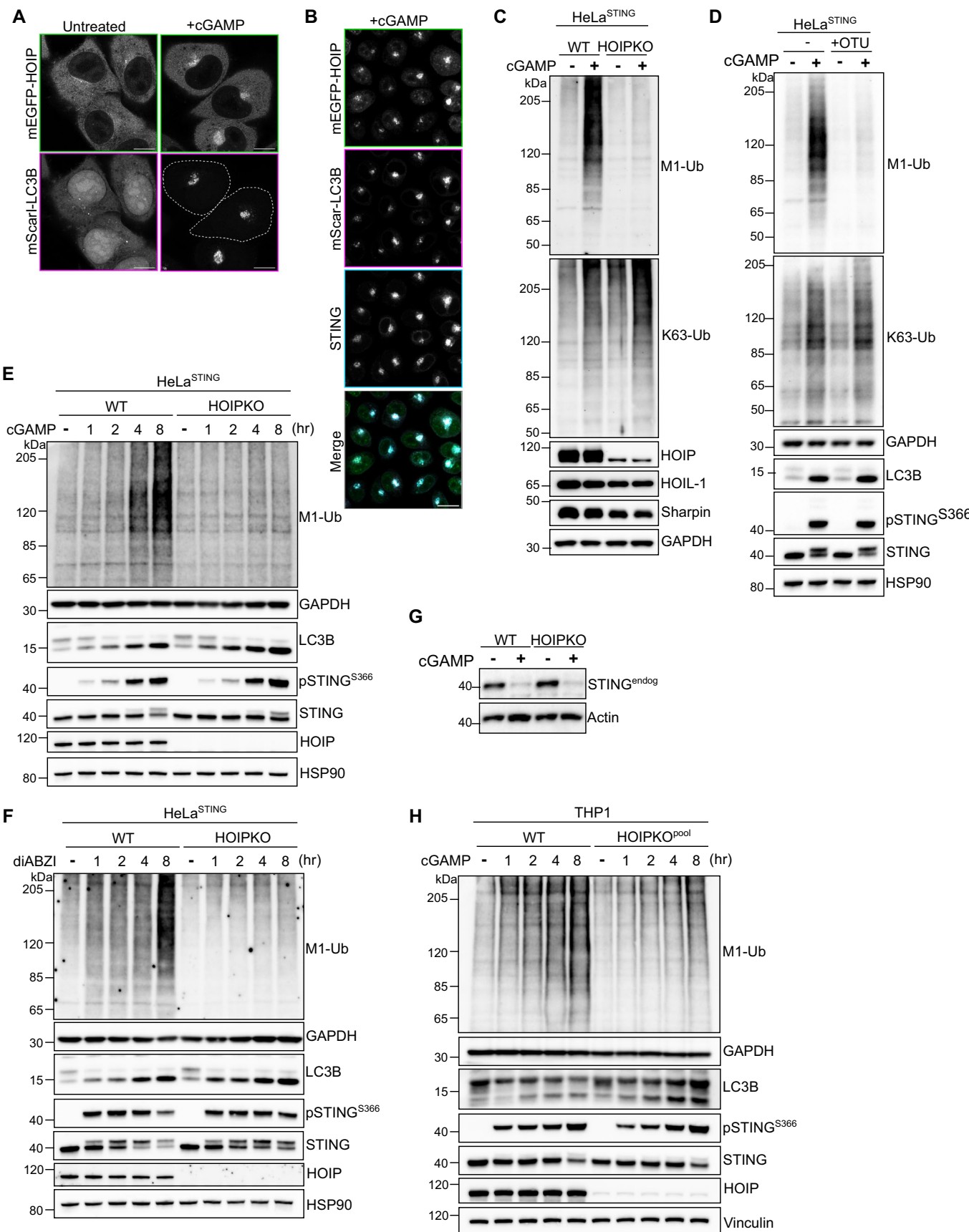

◀ **Figure EV2.   STING activation by multiple agonists induces HOIP-mediated M1 ubiquitin chain formation, which is not required for STING degradation or LC3B lipidation in HeLa or THP1 cells.**

(A) Representative Airyscan-processed confocal images of HeLa^STING cells stably expressing mScarletI-LC3B and mEGFP-HOIP treated with 120 μg/mL of cGAMP for 8 h with no saponin extraction prior to PFA-fixation. Scale bar = 10 μm. Corresponding to representative images in Fig. 2A. Imaging was replicated in three independent experiments. (B) Representative spinning disk confocal images of HeLa^STING; mEGFP-HOIP (green); mScarletI-LC3B (magenta) cells treated with 120 μg/mL of cGAMP for 8 h prior to prior to saponin extraction and PFA-fixation. Scale bar = 20 μm. Corresponding to quantification in Fig. 2B. (C, D) Representative immunoblots of indicated proteins detected in cell lysates from HeLa^STING WT and HOIPKO cells (C) or HeLa^STING and HeLa^STING stably expressing mEGFP-OTULIN (D) prepared following treatment with 120 μg/mL cGAMP for 8 h. Immunoblotting was replicated in three independent experiments. (E, F) Representative immunoblots of indicated proteins detected in HeLa^STING cell lysates from WT and HOIPKO cells prepared following treatment with 120 μg/mL cGAMP (E) or 1 μM diABZI (F) for 1, 2, 4, and 8 h. Immunoblotting was replicated in three independent experiments. (G) Representative immunoblots of endogenous STING were detected in cell lysates from WT and HOIPKO HeLa cells without stable overexpression of STING. Cells were treated with 15 μg/mL cGAMP for 8 h. Immunoblotting was replicated in three independent experiments. (H) Representative immunoblots of indicated proteins detected in THP1 cell lysates from WT and HOIPKO^pool cells prepared following treatment with 120 μg/mL cGAMP for 1, 2, 4, and 8 h. Immunoblotting was replicated in three independent experiments.

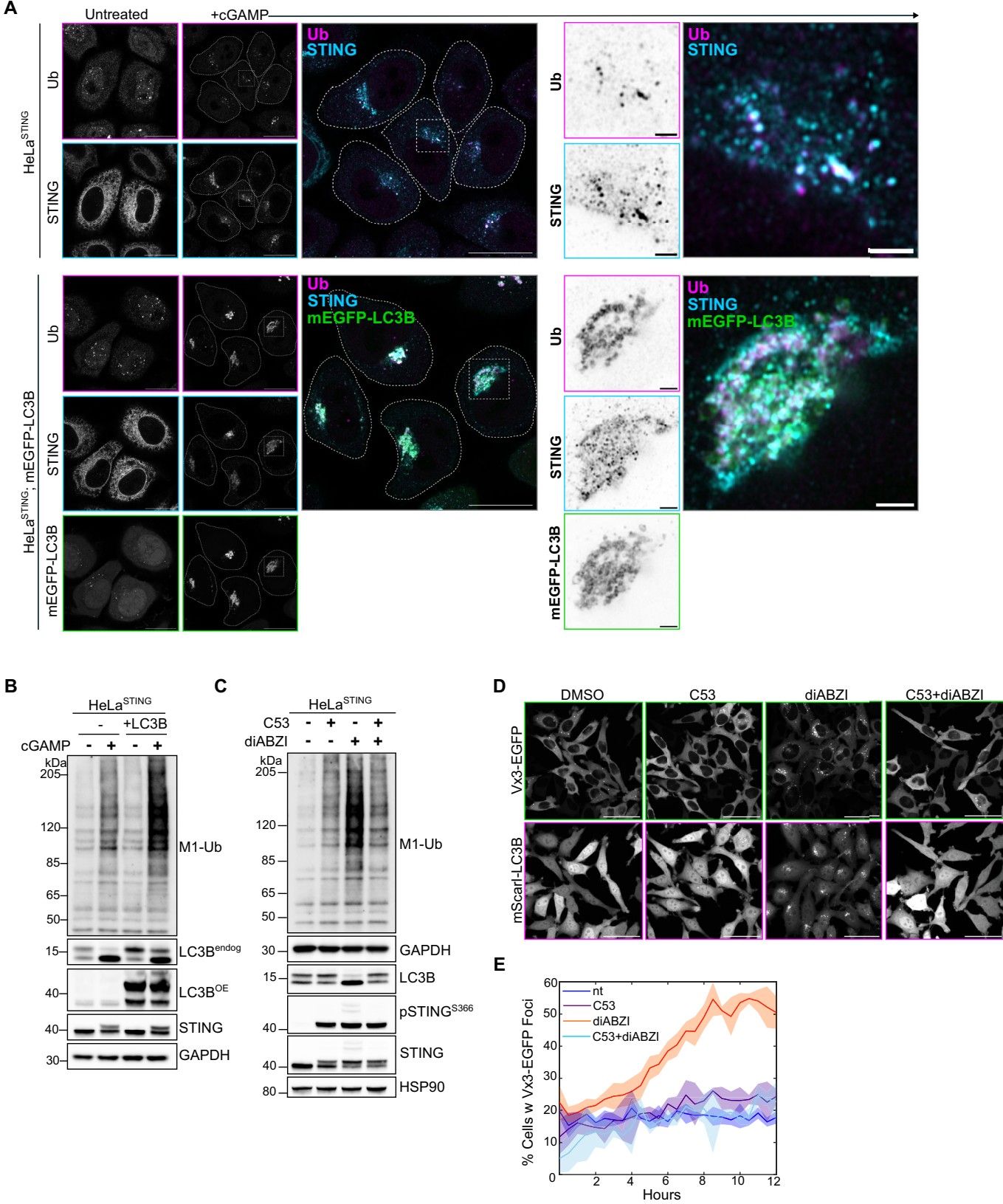

◀ **Figure EV3. Overexpression of LC3B affects STING activation-induced ubiquitylation, which may involve the Golgi neutralizing function of STING.**

(A) Representative Airyscan-processed confocal images of HeLa[STING] and HeLa[STING] cells with stable overexpression of mEGFP-LC3B (green) treated with 120 µg/mL of cGAMP for 8 h prior to PFA-fixation and immunostaining with antibodies raised against mono- and poly-ubiquitin chains (Ub; magenta) and STING (cyan). Scale bar = 20 and 2 µm (inset). Imaging was replicated in two independent experiments. (B) Representative immunoblots of indicated proteins detected lysates from HeLa[STING] WT and HeLa[STING] with stable overexpression of mEGFP-LC3B prepared after treatment with 120 µg/mL cGAMP for 8 h. Immunoblotting was replicated in 3 independent experiments. (C) Representative immunoblots of indicated proteins detected in HeLa[STING] cell lysates prepared after treatment with either DMSO, 10 µM C53, 1 µM diABZI, or both C53 and diABZI for 4 h. Immunoblotting was replicated in three independent experiments. (D, E) Representative spinning disk confocal images of FRT/TREX HeLa cells stably expressing FRT/TO-DD-Vx3-EGFP, BFP-P2A-STING, and mScarletI-LC3B at the 6-hour timepoint following treatment (D) and quantification of the percentage (%) of cells positive for Vx3-EGFP foci over time (E). Cells were incubated with 1 µg/mL Doxycycline and 500 nM Shield1 for 24 h prior to treatment with either DMSO, 10 µM C53, 1 µM diABZI, or both C53 and diABZI, and imaging every 30 min for 12 h on a spinning disk confocal microscope. Scale bar = 50 µm. Quantification is from three wells analyzed in the same experiment. Imaging was replicated in two independent experiments.

 

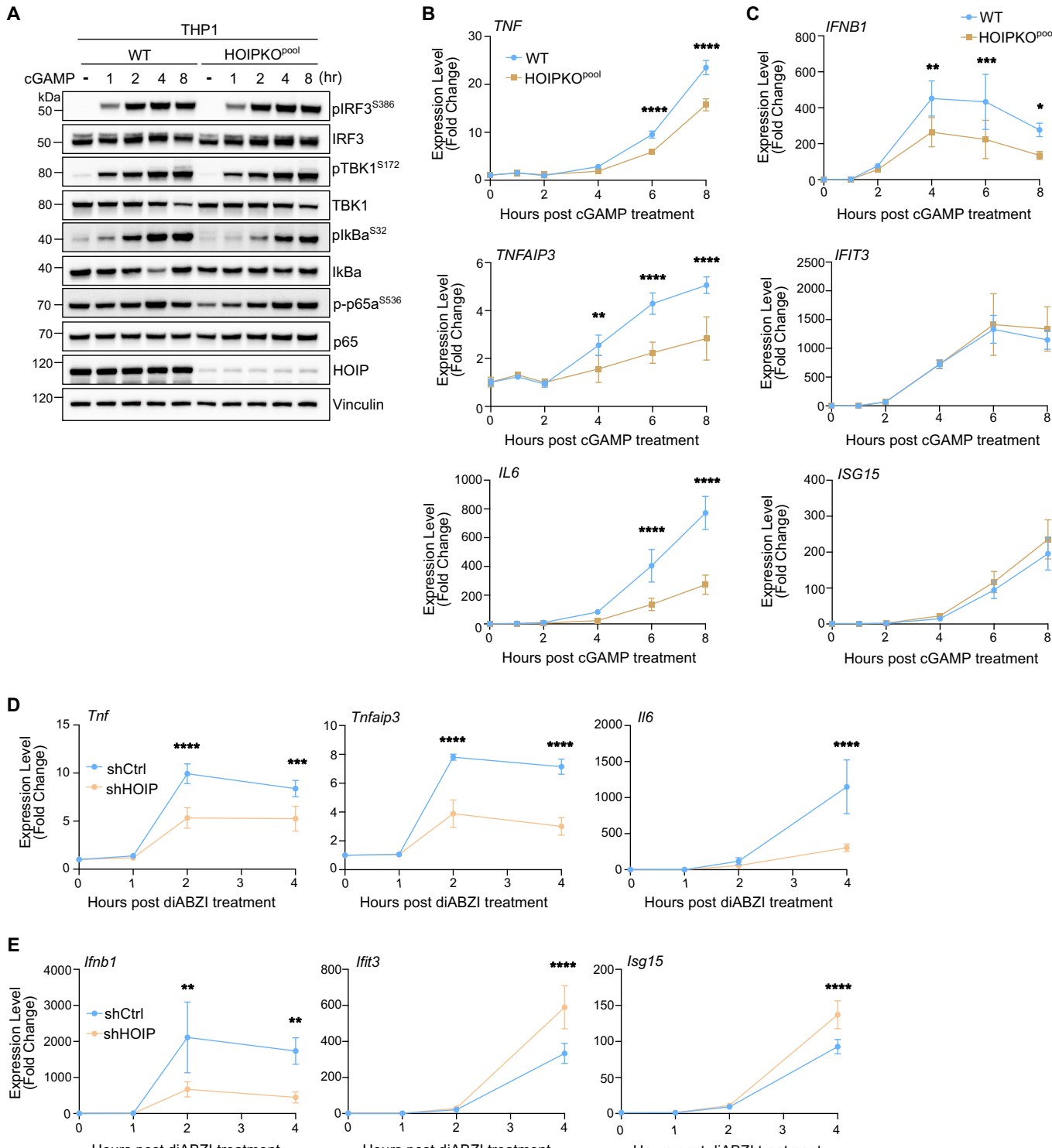

**Figure EV4.   HOIP is important for NFκB signaling following STING activation by cGAMP and in mouse bone marrow-derived macrophages.**

(A) Representative immunoblots of indicated proteins detected in THP1 cell lysates from WT and HOIPKO$^{pool}$ cells prepared following treatment with 120 µg/mL cGAMP for 1, 2, 4, and 8 h. Immunoblotting was replicated in three independent experiments. (B, C) Relative expression changes of indicated NFκB- (B) and IRF3/interferon-related (C) genes detected by quantitative RT-PCR in WT and HOIPKO$^{pool}$ THP1 cells treated with 120 µg/mL cGAMP for 1, 2, 4, and 8 h. Quantification of relative expression is from three independent experiments analyzed at the same time. A two-way ANOVA with Sidak's multiple comparisons test was performed on $2^{-\Delta\Delta Ct}$ values. Mean ± s.d. $n = 3$ *<0.05, **<0.01, ***<0.001, ****<0.0001 (*TNF* 6 h $p$ = <0.0001, 8 h $p$ = <0.0001; *TNFAIP3* 4 h $p$ = 0.0084, 6 h $p$ = <0.0001, 8 h $p$ = <0.0001; *IL6* 6 h $p$ = <0.0001, 8 h $p$ = <0.0001; *IFNB1* 2 h $p$ = 0.0019, 4 h $p$ = 0.0005, 8 h $p$ = 0.0281). (D, E) Relative expression changes of indicated NFκB- (D) and IRF3/interferon-related (E) genes detected by quantitative RT-PCR in WT and shCtrl and shHOIP iBMDM cells treated with 0.2 µM diABZI for 1, 2, and 4 h. Quantification of relative expression is from three independent experiments analyzed at the same time. A two-way ANOVA with Sidak's multiple comparisons test was performed on $2^{-\Delta\Delta Ct}$ values. Mean ± s.d. $n = 3$ **<0.01, ***<0.001, ****<0.0001 (*Tnf* 2 h $p$ = <0.0001, 4 h $p$ = 0.0005; *Tnfaip3* 2 h $p$ = <0.0001, 4 h $p$ = <0.0001; *Il6* 4 h $p$ = <0.0001; *Ifnb1* 2 h $p$ = 0.0011, 4 h $p$ = 0.0031; *Ifit3* 4 h $p$ = <0.0001; *Isg15* 4 h $p$ = <0.0001).

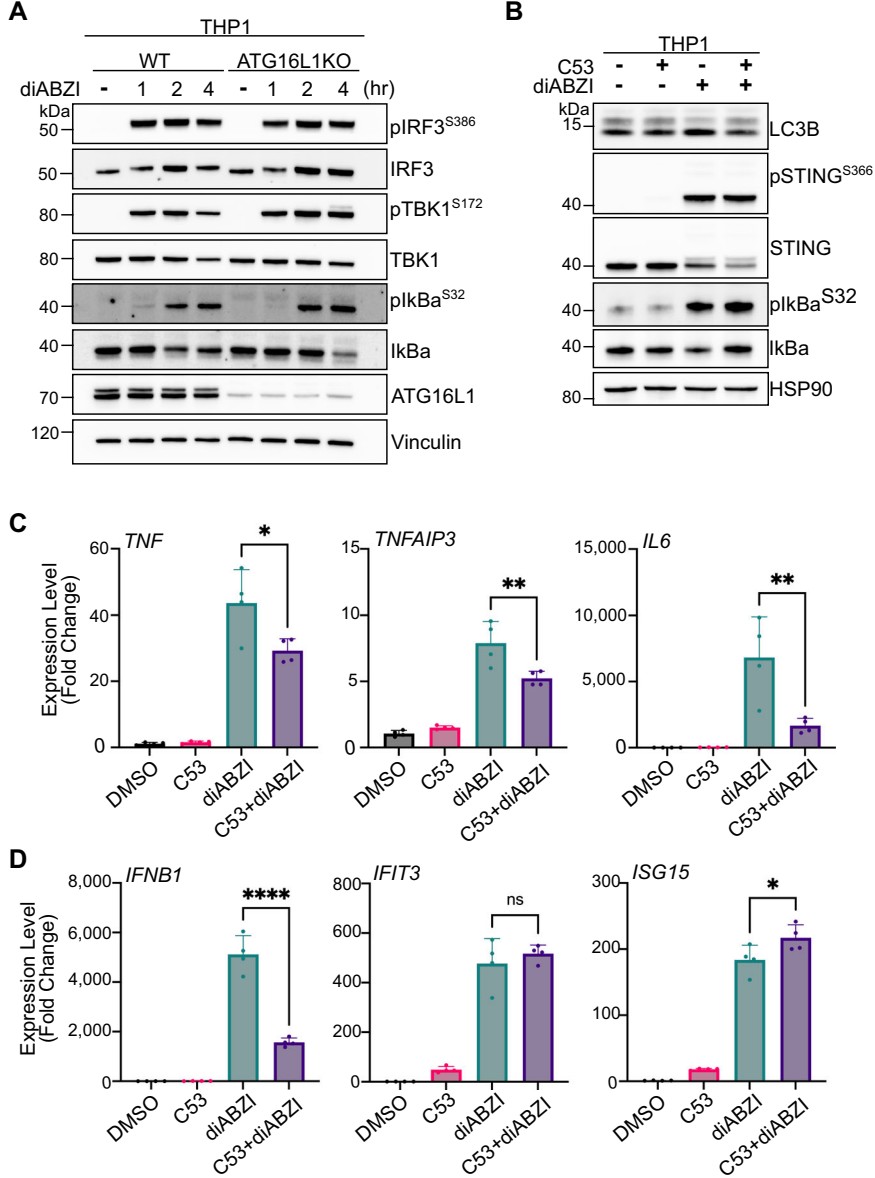

**Figure EV5.  ATG16L1 is not required for IRF3- and NFκB-related signaling induced by STING activation, however the upstream Golgi neutralizing function of STING may affect immune-related gene expression.**

(**A**) Representative immunoblots of indicated proteins detected in THP1 cell lysates from WT and ATG16L1KO cells prepared following treatment with 1 μM diABZI for 1, 2, and 4 h. Immunoblotting was replicated in three independent experiments. (**B**) Representative immunoblots of indicated proteins detected in lysates from WT THP1 cells prepared following treatment with either DMSO, 10 μM C53, 1 μM diABZI, or both C53 and diABZI for 4 h. Immunoblotting was replicated in three independent experiments. (**C, D**) Relative expression changes of indicated NFκB-related genes (**C**) and interferon-related genes (**D**) detected by quantitative RT-PCR in THP1 cells treated with DMSO, 10 μM C53, 1 μM diABZI, or both C53 and diABZI for 4 h. Quantification of relative expression is from four independent experiments analyzed at the same time. A one-way ANOVA with Tukey's multiple comparisons test was performed on $2^{-\Delta\Delta Ct}$ values. Mean ± s.d. $n = 4$ *<0.05, **<0.01, ****<0.0001 (*TNF* $p = 0.0123$; *TNFAIP3* $p = 0.005$; *IL6* $p = 0.0027$; *IFNB1* $p = <0.0001$; *ISG15* $p = 0.0351$).

