## [Peer Review File · The EMBO Journal]

STING induces HOIP-mediated synthesis of M1 ubiquitin chains to stimulate NF- κ B signaling

Tara Fischer, Eric Bunker, Peng-Peng Zhu, Francois Le Guerroue, Mahan Hadjian, Eunice Dominguez-Martin, Francesco Scavone, Robert Cohen, Tingting Yao, Yan Wang, Achim Werner, and Richard Youle

Corresponding author(s): Richard Youle (YouleR@ninds.nih.gov), Richard Youle (YouleR@ninds.nih.gov), Tara Fischer (fischertad@nih.gov)

Review Timeline:

Submission Date:	24th Oct 23
Editorial Decision:	8th Dec 23
Revision Received:	14th Sep 24
Editorial Decision:	17th Oct 24
Revision Received:	20th Oct 24
Accepted:	21st Oct 24

Editor: William Teale

Transaction Report:

Dear Tara and Richard,

Thank you for submitting your manuscript for consideration by the EMBO Journal. It has now been seen by three referees whose comments are shown below.

As we discussed in person, I would like to invite you to submit a revised version of the manuscript, addressing the comments of all three reviewers. I should add that it is EMBO Journal policy to allow only a single round of revision, and acceptance of your manuscript will therefore depend on the completeness of your responses in this revised version.

Thank you for the opportunity to consider your work for publication. I look forward to your revision.
Best wishes,

William

William Teale, PhD
Editor
The EMBO Journal
w.teale@embojournal.org

When submitting your revised manuscript, please carefully review the instructions below and include the following items:

- 1) a .docx formatted version of the manuscript text (including legends for main figures, EV figures and tables). Please make sure that the changes are highlighted to be clearly visible.
- 2) individual production quality figure files as .eps, .tif, .jpg (one file per figure).
- 3) a .docx formatted letter INCLUDING the reviewers' reports and your detailed point-by-point response to their comments. As part of the EMBO Press transparent editorial process, the point-by-point response is part of the Review Process File (RPF), which will be published alongside your paper.
- 4) a complete author checklist, which you can download from our author guidelines ([https://wol-prod-cdn.literatumonline.com/pb-assets/embo-site/Author Checklist%20-%20EMBO%20J-1561436015657.xlsx](https://wol-prod-cdn.literatumonline.com/pb-assets/embo-site/Author%20Checklist%20-%20EMBO%20J-1561436015657.xlsx)). Please insert information in the checklist that is also reflected in the manuscript. The completed author checklist will also be part of the RPF.
- 5) Please note that all corresponding authors are required to supply an ORCID ID for their name upon submission of a revised manuscript.
- 6) We require a 'Data Availability' section after the Materials and Methods. Before submitting your revision, primary datasets produced in this study need to be deposited in an appropriate public database, and the accession numbers and database listed under 'Data Availability'. Please remember to provide a reviewer password if the datasets are not yet public (see <https://www.embopress.org/page/journal/14602075/authorguide#datadeposition>). If no data deposition in external databases is needed for this paper, please then state in this section: This study includes no data deposited in external repositories. Note that the Data Availability Section is restricted to new primary data that are part of this study.

Note - All links should resolve to a page where the data can be accessed.

- 7) When assembling figures, please refer to our figure preparation guideline in order to ensure proper formatting and readability in print as well as on screen:
<http://bit.ly/EMBOPressFigurePreparationGuideline>

8) For data quantification: please specify the name of the statistical test used to generate error bars and P values, the number (n) of independent experiments (specify technical or biological replicates) underlying each data point and the test used to calculate p-values in each figure legend. The figure legends should contain a basic description of n, P and the test applied. Graphs must include a description of the bars and the error bars (s.d., s.e.m.).

9) We would also encourage you to include the source data for figure panels that show essential data. Numerical data can be provided as individual .xls or .csv files (including a tab describing the data). For 'blots' or microscopy, uncropped images should be submitted (using a zip archive or a single pdf per main figure if multiple images need to be supplied for one panel). Additional information on source data and instruction on how to label the files are available at .

10) We replaced Supplementary Information with Expanded View (EV) Figures and Tables that are collapsible/expandable online (see examples in <https://www.embopress.org/doi/10.15252/emboj.201695874>). A maximum of 5 EV Figures can be typeset. EV Figures should be cited as 'Figure EV1, Figure EV2' etc. in the text and their respective legends should be included in the main text after the legends of regular figures.

12) Our journal encourages inclusion of *data citations in the reference list* to directly cite datasets that were re-used and obtained from public databases. Data citations in the article text are distinct from normal bibliographical citations and should directly link to the database records from which the data can be accessed. In the main text, data citations are formatted as follows: "Data ref: Smith et al, 2001" or "Data ref: NCBI Sequence Read Archive PRJNA342805, 2017". In the Reference list, data citations must be labeled with "[DATASET]". A data reference must provide the database name, accession number/identifiers and a resolvable link to the landing page from which the data can be accessed at the end of the reference. Further instructions are available at .

Further instructions for preparing your revised manuscript:

We realize that it is difficult to revise to a specific deadline. In the interest of protecting the conceptual advance provided by the work, we recommend a revision within 3 months (7th Mar 2024). Please discuss the revision progress ahead of this time with the editor if you require more time to complete the revisions. Use the link below to submit your revision:

Referee #1:

In the manuscript "STING induces LUBAC-mediated synthesis of linear ubiquitin chains to stimulate innate immune signalling", the authors build upon the existing connection between STING activation and LC3B lipidation and further add the synthesis of M1-linked ubiquitin chains and clustering of ubiquitin adaptors/receptors and their importance for NF κ B activation and cytokine production. In a simplified view, the manuscript can be divided into two halves. The first half focuses on the STING activation using cGAMP and the ensuing synthesis of linear ubiquitin, that is mediated by LUBAC complex, and recognized by TNIP1 and OPTN. The second part focuses on the transcriptional consequences of the synthesis of the M1-linked ubiquitin chains after STING activation, translocation and activity in THP-1 cells and HeLa cells.

The first half is well characterized and there are only a few controls and clarifications missing. The second part makes the manuscript a bit more complicated to follow because the authors change the inducers of STING activation (cGAMP to TNF and diAZBI) and use a different cell line (THP-1 cells, instead of HeLa cells) for one set of experiments, which leads to interesting but confusing results. For example, on HeLa cells, TNF α expression is dependent on M1-linked ubiquitin chains after TNF α stimulation and treatment with STING agonist diAZBI but not after cGAMP stimulation. The authors should characterize why cGAMP and diAZBI have different downstream effects on cytokine production if they trigger the same pathway (STING, M1-linked ubiquitin chains, TNIP1 and OPTN clustering) and address the following concerns:

Major comments:

- 1) One of the critical aspects of this work is the localization of the M1-linked ubiquitin chains to LC3B- and STING-positive foci. Given that STING can be translocated from the ER to the Golgi, where are these foci located? Are they on the Golgi or the ER, or on both?
- 2) Also, the authors show on Figure 1A a beautiful image of Ub-LC3B-STING foci, but they do not apply the same quantification that they performed for Ub-LC3B foci (Figure S1). They should extend the quantifications that were performed in Figure S1 for STING⁺ foci, LC3B⁺-STING⁺, Ub⁺-STING⁺, and LC3B⁺-Ub⁺-STING⁺ foci. Also, the authors should perform these analyses for HOIP, TNIP1, OPTN. They develop a very interesting analysis pipeline for Figure 1 and S1 and then simply show the co-localization coefficients for the remaining manuscript.
- 3) Are the M1-linked chains of ubiquitin anchored to any protein or unanchored?
- 4) The connection between OPTN and the pathway is not convincing. In Figure 3, there are intense foci of OPTN close to LC3B, but there is much more OPTN elsewhere. What is the proportion of OPTN in the LC3B foci, when compared to the total OPTN present in the cells?
- 5) Is TNF α a good stimulus in this context? TNFR stimulation leads directly to the synthesis of M1-linked ubiquitin chains at the plasma membrane, correct? In that sense, STING activation and consequent M1-linked ubiquitin chain synthesis in its vicinity would be a slower and possibly dispensable mechanism of NF κ B activation. Did the authors test TNF α activation together with C53, to assess the contribution of STING-induced M1-linked chains on immune signaling?
- 6) The experiments in Figure 4 add little to the paper and are in need of controls. In THP-1 cells after diAZBI stimulation, are the M1-linked ubiquitin chains being synthesized in LC3B-STING foci? In THP-1 cells, does cGAMP also induce TNF α expression independently of HOIP? The author may consider changing this into a Supplementary Figure and "promoting" Supplementary Figure 5 into a Main Figure.
- 7) The biggest conceptual problem of the manuscript is the fact that cGAMP and diAZBI have different downstream effects (especially for IFN β and IL-6) when they bring about the same aggregation of LC3B, STING and Ub into the same foci. From comparing the images in Figure 1 (cGAMP activation) and Figure S7 (diAZBI activation), it seems that cGAMP induces a bigger aggregation of STING than diAZBI. Could this be a reason for the opposing effects? Could the STING-induced M1-linked ubiquitin chains actually be inhibiting the immune response? Are the proportions of OPTN and TNIP1 different between the two stimuli? The authors should also test the effect of diAZBI treatment on HeLa cells that are KO for HOIP and that overexpress OTU. This would make it easier to assess which pathway really depend on M1-linked ubiquitin chains synthesized following STING activation.

Minor comments:

- 8) It would be useful if the authors could provide some schematic representation of the MATLAB analysis pipelines they used for the readers to understand.
- 9) The authors should show merged images, especially for the Supplementary Figures, where there is no space limitation and makes it easier to match the foci.
- 10) In Figure 1A, why does the ubiquitin signal look so punctuate in the untreated conditions?
- 11) How big was the dilation of the foci to detect ubiquitin? What is the size of the "doughnut", as the authors put it?
- 12) Did the authors look for other autophagy/ubiquitin receptors, like p62 or NDP52, in close proximity with STING and LC3B after cGAMP treatment?
- 13) In Figure S4, OPTN-GFP seems overexposed on some cells. Is this just to increase contrast to the Figures, or was the analysis performed only on cells where OPTN-GFP was not overexposed?
- 14) What is the effect of knocking-out or knocking-down the TNIP1 and OPTN for immune signalling and cytokine expression?

Referee #2:

This manuscript focuses on the interconnections of LC3B and linear ubiquitin (M1-Ub) chains in the control of downstream innate immune signaling. The manuscript expands on previous work from the authors describing a STING-activation non-canonical autophagy pathway. They show that the activation of STING results in increased K63- and M1- ubiquitin chain formation at LC3B-associated Golgi membranes. Although STING activation induces LC3B lipidation by a non-canonical autophagy pathway, the authors show this pathway has distinct features from the most well-studied non-canonical autophagy pathway, called CASM. Specifically, while both STING and the monensin-induced CASM pathways result in LC3B lipidation as well as increased K63-Ub chain formation, only the STING pathway is able to induce robust linear ubiquitin chain formation. The authors further demonstrate that M1-Ub correlates with the activation of NFkB and IRF3-mediated signaling in THP1 monocytes. Overall the experiments are interesting but preliminary in scope. Importantly, the molecular relationships between STING-induced non-canonical autophagy and linear Ub chain formation remain obscure and the effects of M1-Ub on innate immune signaling are limited to a single cell line. Significant additional work is necessary to support the authors main conclusions that LUBAC synthesis of linear ubiquitin chains regulates STING-mediated innate immune signaling.

- 1) Although the studies demonstrate differences between STING activation and other non-canonical pathways with regard to the production of M1-Ub chains, additional insight into the mechanisms underlying linear Ub chain formation at LC3 positive membranes is needed. Is LC3B conjugation functionally required for HOIP/LUBAC recruitment to STING foci in response to cGAMP?
- 2) The studies have been almost entirely been conducted in HELA cells expressing STING, except for the one key set of experiments in THP1 looking at immune signaling. Notably, the downstream effects on NFkB target expression are observed in THP1 cells, but not in HELA-STING cells, at least based on the data provided in Supplementary Fig 6. This casts doubt on the broad physiological role of M1-Ub in the control of NFkB and IRF3 signaling, which is the major conclusion of the paper. Additional evidence in other cell lines or tissue types is necessary to more robustly support the conclusions that have been drawn.
- 3) For the immunoblotting throughout the studies, the authors show single blots with no attempt as quantification. Quantification of multiple independent experiments with statistical analysis should be provided.
- 4) The physiological relevance of the recruitment of the autophagy cargo receptors TNIP1 and OPTN to LC3 positive membranes during STING activation remains obscure. These studies seem out of place from the rest of the manuscript.
- 5) The role of LC3B conjugation in M1-Ub is circumstantial in Fig 5. The role of ATGs required for LC3 conjugation such as ATG7 or ATG3 should be directly tested for their effects on M1-Ub chain formation downstream of cGAMP. Furthermore, as discussed above in point 1, does LC3B play any role in the recruitment of HOIP or other LUBAC components. Overall, these studies lack mechanistic depth.

Referee #3:

In the manuscript by Fischer et al., the authors examine the role of LUBAC-mediated M1 linear Ub in STING signalling. They reveal that in addition to K63 Ub (but not K48) activation of STING leads to M1 Ub chain formation associated with LC3B. The authors also demonstrate that STING activation induces the interaction between LC3B and the adaptor proteins, TNIP1 and OPTN, while the deletion of HOIP appears to lead to a loss of STING-dependent cytokine gene expression (in THP-1 cells). Some aspects of STING signalling are not well understood (e.g., NF-kB activation) and here M1 Ub is implicated in STING responses, hence this manuscript is of interest to the field. However, the current data is not always coherent (inconsistency in results from different cell types) and doesn't currently provide enough mechanistic insights into STING signalling other than that M1 Ub is induced in response to STING activation and may be important for cytokine gene expression (in THP1). A major missing piece to this story is also where are the M1 Ub chains are added? Is it directly to STING, or STING complexes, or LC3B alone? This should be examined further.

This manuscript requires major revision before it should be considered further for publication at EMBO J. Below are some specific queries/comments:

1. Introduction: This section should include the findings describing that STING activation also leads to its degradation (which is used as an observation in the manuscript) by the lysosome (see PMID: 29241549) - a process that requires both the ESCRT system and aspects of non-canonical autophagy (see PMID: 36918692 PMID: 36739287 PMID: 37139896). These 3 more recent studies all showed (K63) Ub was also critical for STING degradation and this should be mentioned for the reader and also perhaps written about in the discussion section.
2. Fig.1: A major concern for me was the choice of cell type predominantly used for much of the work. Why did the authors use HeLa cells expressing an untagged version of STING? Why not just examine these responses using the endogenous STING levels in HeLa? I believe HeLa express STING at appropriate levels for activation. If not and they are not function with endogenous STING then they are probably not that appropriate for these studies.
3. Fig.1: The authors should show the reader a comparison between endogenous STING from WT HeLa vs HeLa-STING, showing at least: STING protein expression levels; and upon activation (i.e., P-STING, P-TBK1, P-IRF3).
4. Fig.1a: It would be helpful to also include an enlarged triple colour overlay here. Also, what %age of Ub and LC3B localises to STING, both together and also separately (i.e., STING and Ub, STING and LC3B)?
5. Fig.S1d-f: This figure would be improved by also including staining for STING as per Fig.1a.
6. Fig. 1e and Fig.S1h: The 2 sentences describing these data should be combined so it is clear which data has used which antibody. Also, some supporting data from more relevant primary immune cells would be important. E.g., does activation of primary human monocytes by STING also lead to a clear increase in detection of M1 Ub chains?
7. Fig.1f: I found this data a little unclear. It is shown as Fold Change but over what? This should be clear on the y-axis and mentioned in the text. In Fig.1g the axis should also state FC over untreated. Further, why is Fig.1g seemly presented differently to Fig.1f? This should be consistent: Fig.1g NT and cGAMP - is this 8-fold change statistically significant?
8. Fig.1: I am left wondering here if STING itself (or the signalling complex) is subject to M1 Ub? It is not clear from the current data but this would be important to examine. For instance, STING IP then IB for M1-Ub (and K63, K48 Ubs to show specificity). This should also be undertaken in THP1 monocytes.
9. Fig.2a: Examines HOIP interactions with LC3B. Does HOIP also appear to interact with STING? Also, what about the other LUBAC components HOIL and SHARPIN - do they interact with LC3B and/or STING?
10. Fig.2b: Is the remaining band in the blot not HOIP? If so, are these KO or knock down cells?
11. Fig.2c: Here the HOIP, HOIL and Sharpin blots should also be presented below, as in 2b.
12. Fig.2d: Some STING responses are intact in HOIP KO or OTULIN overexpression: In addition, the authors need to examine p-TBK1, p-IRF3 and p-p65 NF-kB. Kinetics should also be examined not just at 8 h. Are these also intact? This should also be done in HOIP KO THP-1 cells later. Also why is the untagged STING not degraded in these cells after 8h?
13. Fig.2d: Associated to this figure, the data would suggest TBK1 is still actively recruited to STING to mediate STING phosphorylation. Is this the case? Does STING associate with TBK1 via IP?
14. Fig.S3e: Here endogenous STING degradation is examined. In these cells the authors should also assess the readouts p-STING, LC3B lipidation, M1 Ub. It would also be informative to show p-TBK1, p-IRF3 and p-p65 NF-kB.
15. Fig.2: Does M1 Ub chain formation by cGAMP require TBK1 i.e. Are M1 Ub chains still induced in TBK1 KO cells (preferably THP1)?
16. Fig.3a-b: TNIP1 and OPTN are detected at LC3B foci by immunostaining following cGAMP treatment. Here LC3B is overexpressed (as is STING): would this happen under more physiological conditions or endogenous expression levels. Also are TNIP1 and OPTN detected with STING? Or just LC3B?
17. Fig.3: Is there any functional consequences of TNIP1 and/or OPTN to STING responses? Why are these adaptors localising to LC3B? What effect on STING responses does KO of these have? Currently this data adds little to the manuscript and shows no relevance to STING immune responses.
18. Fig.4c: Does loss of HOIP effect STING-induced MAPK or p-p105 NF-kB?
19. Fig 4 and Fig.S5: If find the inconsistency in results here very puzzling. The logic that we are following from the beginning is that: STING activation leads to M1 Ub chains, which is dependent on HOIP, which when KO (only in THP1) reduces IFN and NF-kB cytokines as suggested by the title "STING induces LUBAC-mediated synthesis of linear ubiquitin chains to stimulate innate immune signaling". So why are HeLa not the same? I don't believe the throwaway line "however there may be distinct mechanisms in HeLa cells" is an acceptable answer. Together with the overexpression of untagged STING, this left me strongly questioning the relevance of any findings using the HeLa system.
20. Fig.5 lead in: NLRP3 induces pyroptotic cell death.
21. Fig.5a: Does C53 only induce STING activation in HeLa when STING is overexpressed?
22. Fig.5b: Does C53 induce M1 Ub in WT HeLa with endogenous STING levels?
23. Fig. S8e: In THP1 C53 doesn't induce STING activation but also appears not to induce significant LC3B lipidation. Why? The data is hard to interpret with such poor induction of LC3B lipidation.
24. Fig.5c-d: Why do the ISGs IFIT3 and ISG15 go up with C53 + diABZI treatment? If IFN β goes down, shouldn't they also be reduced?
25. Related to Fig.e-f: What is the effect of C53 on STING signalling generally in THP1? The authors should examine p-TBK1, p-IRF3 and p-p65 NF-kB as well.
26. Fig.5f: Does SopF reduce STING-induced LC3B lipidation examined by WB? If SopF blocks LC3B lipidation generally, shouldn't M1 Ub after cGAMP be reduced in the presence of SopF not unchanged? I find the logic here very unclear. SopF

experiments should also be conducted in THP1 cells.

27. Fig.5 conclusion: "suggests that STING activation induced LC3B lipidation may promote ubiquitin chain formation" but its stated SopF blocks LC3B lipidation. As stated above why doesn't SopF therefore reduce M1 Ub?

28. Discussion: IFN is also affected by HIOP KO. In some cell types NF- κ B contributes to a full IFN β transcriptional response - see the IFN β enhancesome. Hence, without examining p-IRF3 in these studies one cannot concluded where this M1 Ub may be affecting things? Is it just acting on NF- κ B?

29. Discussion: The mention that TBK1 recruitment to STING may involve OPTN should either be investigated or not mentioned.

30. General concern: The authors use either 60 or (more often) 120 μ g/mL of 2'3'-cGAMP - which are very high concentrations (commonly used between 5-20 μ g/mL). It is currently unclear exactly how cells are stimulated with cGAMP. Please explain how this is done. Is cGAMP added to media or transfected? This form of cGAMP is generally transfected while other stabilised forms of 2'3'-cGAMP e.g. 2'3'-cGAM(PS)₂ (Rp/Sp) are added to media at much lower concentrations. Otherwise for human cells one can also use diABZI (as is done in some later aspects of this manuscript. Also, why is diABZI used for THP1 experiments and cGAMP for HeLa? Does diABZI elicit the same results in HeLa and vice versa?

31. Authors state that M1 Ub may be associated with STING-induced LC3B: Use C53 to show no STING activation but no M1 Ub?

Point by point response to reviewers.**Referee 1**

In the manuscript "STING induces LUBAC-mediated synthesis of linear ubiquitin chains to stimulate innate immune signalling", the authors build upon the existing connection between STING activation and LC3B lipidation and further add the synthesis of M1-linked ubiquitin chains and clustering of ubiquitin adaptors/receptors and their importance for NF κ B activation and cytokine production. In a simplified view, the manuscript can be divided into two halves. The first half focuses on the STING activation using cGAMP and the ensuing synthesis of linear ubiquitin, that is mediated by LUBAC complex, and recognized by TNIP1 and OPTN. The second part focuses on the transcriptional consequences of the synthesis of the M1-linked ubiquitin chains after STING activation, translocation and activity in THP-1 cells and HeLa cells. The first half is well characterized and there are only a few controls and clarifications missing.

The second part makes the manuscript a bit more complicated to follow because the authors change the inducers of STING activation (cGAMP to TNF and diABZI) and use a different cell line (THP-1 cells, instead of HeLa cells) for one set of experiments, which leads to interesting but confusing results. For example, on HeLa cells, TNF α expression is dependent on M1-linked ubiquitin chains after TNF α stimulation and treatment with STING agonist diABZI but not after cGAMP stimulation. The authors should characterize if they trigger the same pathway (STING, M1-linked ubiquitin chains, TNIP1 and OPTN clustering) and address the following concerns:

We appreciate the reviewer comments and have addressed them as follows.

Major comments:

1) One of the critical aspects of this work is the localization of the M1-linked ubiquitin chains to LC3B- and STING-positive foci. Given that STING can be translocated from the ER to the Golgi, where are these foci located? Are they on the Golgi or the ER, or on both?

We agree the localization of M1 chains relative to LC3B foci and STING is an interesting question to address. Despite attempting several methods, we were unable to directly identify the spatial localization of M1-Ub chains due to limitations of cross-reactivity of the M1-Ub antibody, which has previously been reported (1, 2), and issues with published reporters. Indirectly, we now quantitatively detect the spatial co-localization of STING, ubiquitin, HOIP, and LC3B in perinuclear foci using both cGAMP and diABZI treatment (**Fig. 1-3 and EV1-3**). As the reviewer notes, STING is well documented to traffic from the ER to the Golgi following both cGAMP and diABZI treatment by spatial co-localization with ER and Golgi markers (3–7), as well as proximity-labeling studies detecting STING in proximity to ER proteins when unstimulated and then proteins related to various Golgi compartments upon activation and trafficking (8, 9). We and others have demonstrated that STING activation induced LC3B foci are associated with small vesicles near the Golgi detected by correlative light electron microscopy (6, 10, 11). After cGAMP or diABZI treatment, STING-induced LC3B foci co-localize with Golgi markers by immunofluorescence (6) and with fluorescent Golgi reporters (11). Our analysis indicates that HOIP and ubiquitin colocalize with LC3B and STING in the Golgi-localized, perinuclear region of cells (**Fig. 1-3 and EV1-3**). Collectively, these data strongly suggest that M1-Ub, HOIP, LC3B, and STING are all localized at the Golgi. We also include an acknowledgement of this limitation in our study in the discussion, **lines 246-250**.

2) Also, the authors show on Figure 1A a beautiful image of Ub-LC3B-STING foci, but they do not apply the same quantification that they performed for Ub-LC3B foci (Figure S1). They should extend the quantifications that were performed in Figure S1 for STING+ foci, LC3B+-STING+, Ub+-STING+, and LC3B+-Ub+-STING+ foci. Also, the authors should perform these analyses for HOIP, TNIP1, OPTN. They develop a very interesting analysis pipeline for Figure 1 and S1 and then simply show the co-localization coefficients for the remaining manuscript.

Thank you for this important point. We have now included new data detecting the localization of STING related to LC3B and Ubiquitin foci following both cGAMP and diABZI stimulation of STING (**Fig. 1 and EV1**) with

quantification of STING+ and STING+Ub+ LC3B foci, as well as STING+ and STING+LC3B+ Ub foci. We provide the same analysis with mEGFP-HOIP in **Fig. 2 and EV2**. We discuss TNIP1 and OPTN in detail below.

3) Are the M1-linked chains of ubiquitin anchored to any protein or unanchored?

That is an interesting issue. As K63-linked ubiquitin chain tools are more widely available compared to M1 ubiquitin chain tools, we performed IP Mass Spec to address this issue. We found the K63-specific TUBEs pull down many different proteins, 80% of which are standard Golgi localized proteins (data is planned for a future manuscript). Also, in the MyD88 pathway, M1 Ubiquitin chains are known to be anchored to K63 Ubiquitin chains, called mixed chains, and other proteins (12). Thus, most of the M1 chains localized to the LC3B foci would be likely attached to proteins. Exploring the significance of the protein platforms for the ubiquitin chains would be a very large endeavor. In some cases, the proteins linked to ubiquitin are also less important than the subcellular localization of the platform and the ubiquitin binding proteins which are recruited to the platform are more important in downstream signaling cascades. However, identifying the proteins ubiquitylated downstream of STING and HOIP is important and is a future aim.

4) The connection between OPTN and the pathway is not convincing. In Figure 3, there are intense foci of OPTN close to LC3B, but there is much more OPTN elsewhere. What is the proportion of OPTN in the LC3B foci, when compared to the total OPTN present in the cells?

We agree that it appears not all OPTN go to the foci, either with detection of endogenous OPTN by IF or with overexpression, however it is difficult to determine how much normal background from IF staining is contributing to the cytosolic signal. OPTN is one of 2 proteins (with TNIP1) that has both an LC3-interacting region (LIR) and a linear ubiquitin binding domain (UBAN). As we found that M1 ubiquitin is localized at LC3B foci, we examined whether OPTN may be localized there and whether its localization was dependent on either LIR, the UBAN, or both. We agree that the functional consequences of OPTN is not yet known, but we feel exploring this is beyond the scope of the manuscript. Given this, and other reviewer comments regarding the TNIP1 and OPTN data, as well as the substantial new results we have added to address the other reviewer comments, we ask to delete these minor and confirmatory results.

5) Is TNF α a good stimulus in this context? TNFR stimulation leads directly to the synthesis of M1-linked ubiquitin chains at the plasma membrane, correct? In that sense, STING activation and consequent M1-linked ubiquitin chain synthesis in its vicinity would be a slower and possibly dispensable mechanism of NF κ B activation. Did the authors test TNF α activation together with C53, to assess the contribution of STING-induced M1-linked chains on immune signaling?

TNF α is well understood to induce NF κ B signaling that is partially dependent on M1 ubiquitin chain formation (13, 14). We used TNF α treatment only as a positive control to demonstrate that we find similar effects of M1 ubiquitin chain formation on NF κ B signaling in HOIPKO and depleted conditions, as previously reported. We do not claim or suggest any role of STING in TNF α treatment and apologize for this confusion. We realize this may be confusing as TNF α is a transcriptionally upregulated target of STING. The TNF α data is now exclusively related only to HeLa cell experiments and moved to the supplemental figure 1. We have clarified the use of TNF α as a positive control in the figure legend.

6) The experiments in Figure 4 add little to the paper and are in need of controls. In THP-1 cells after diABZI stimulation, are the M1-linked ubiquitin chains being synthesized in LC3B-STING foci? In THP-1 cells, does cGAMP also induce TNF α expression independently of HOIP? The author may consider changing this into a Supplementary Figure and "promoting" Supplementary Figure 5 into a Main Figure.

We think Figure 4 is an important point of the manuscript as it demonstrates the biological relevance of M1 ubiquitin chain formation induced by STING activation. In new **Figure 2**, we show that HOIP dependent M1 ubiquitin chains in THP1 cells are synthesized upon STING activation similarly to HeLa, but we see the reviewer's point questioning whether the spatial localization is the same as HeLa. Unfortunately, THP1 cells are cultured in suspension and poor for imaging but are more ideal than HeLa for testing immune signaling, as they are myeloid derived. Although we attempted to 'flatten' THP1 cells onto glass coverslips by centrifugation,

the size of the cells was too small to determine any specific spatial information in the cytosolic region. Thus, determining the spatial localization of Ub, LC3B, HOIP, and STING in THP1 cells is technically implausible with our available tools. However, we were able to address the relationship of LC3B lipidation and M1 Ub chain formation genetically, showing that ATG16L1KO in both HeLa (**Fig. 3**) and THP1 (**Fig. 5 and EV5**) is not required for M1-Ub formation or for STING-dependent immune signaling. For the second question, we now include new data in **Fig. 4 and EV4** showing that both diABZI and cGAMP activation of STING induces HOIP-dependent *TNF* gene expression in THP1 cells at several time points.

7) The biggest conceptual problem of the manuscript is the fact that cGAMP and diABZI have different downstream effects (especially for IFN β and IL-6) when they bring about the same aggregation of LC3B, STING and Ub into the same foci. From comparing the images in Figure 1 (cGAMP activation) and Figure S7 (diAZBI activation), it seems that cGAMP induces a bigger aggregation of STING than diAZBI. Could this be a reason for the opposing effects? Could the STING-induced M1-linked ubiquitin chains actually be inhibiting the immune response? Are the proportions of OPTN and TNIP1 different between the two stimuli? The authors should also test the effect of diAZBI treatment on HeLa cells that are KO for HOIP and that overexpress OTU. This would make it easier to assess which pathway really depend on M1-linked ubiquitin chains synthesized following STING activation.

cGAMP and diABZI are both STING agonists that activate STING signaling (15). However, cGAMP and diABZI have different temporal kinetics, as cGAMP requires transporters through the plasma membrane (16, 17) and diABZI is membrane permeable (15). We used these two agonists to induce STING activation for different experiments throughout the manuscript, and in different cell lines, and acknowledge how this has caused confusion in the interpretation of results. We now include data for both cGAMP and diABZI treatment in several new experiments (**Fig. EV1, Fig. 2 and EV2, Fig. 3, Fig. 4, and EV4**), including data at multiple time points to demonstrate the differences in temporal kinetics. Importantly, we also include data detecting HOIP-dependent gene expression over time for both cGAMP and diABZI treatment across several time points in THP1 cells (**Fig. 4 and EV4**). Additionally, cGAMP and diABZI both induce similar levels of co-localization between STING, Ub, and LC3B (**Fig. 1 and EV1**). Therefore, our data show that both agonists consistently produce similar results in all assays, albeit with differing temporal kinetics as previously reported (15).

Minor comments:

8) It would be useful if the authors could provide some schematic representation of the MATLAB analysis pipelines they used for the readers to understand.

We have provided a MATLAB schematic in Supplemental Figure 2.

9) The authors should show merged images, especially for the Supplementary Figures, where there is no space limitation and makes it easier to match the foci.

We have included merged panels in most figures.

10) In Figure 1A, why does the ubiquitin signal look so punctuate in the untreated conditions?

Ubiquitin localizes to puncta in the nucleus at steady state, as is known. This nuclear location is clearly distinct from the cytosolic localization at the Golgi after STING activation.

11) How big was the dilation of the foci to detect ubiquitin? What is the size of the "doughnut", as the authors put it?

They were dilated out 10 pixels which equals 1.6 μm in each direction but excluding other punctae/foci, other cells, and the space between cells.

12) Did the authors look for other autophagy/ubiquitin receptors, like p62 or NDP52, in close proximity with STING and LC3B after cGAMP treatment?

NDP52 and p62 are quite nonspecific ubiquitin binding proteins and may also be localized there. As the functional significance of autophagy receptors binding the foci is unknown and leads to confusion, we ask to delete the TNP1 and OPTN data.

13) In Figure S4, OPTN-GFP seems overexposed on some cells. Is this just to increase contrast to the Figures, or was the analysis performed only on cells where OPTN-GFP was not overexposed?

Please see our comment to Point 12 above.

14) What is the effect of knocking-out or knocking-down the TNIP1 and OPTN for immune signalling and cytokine expression?

Please see our comment to Point 12 above.

Referee 2

This manuscript focuses on the interconnections of LC3B and linear ubiquitin (M1-Ub) chains in the control of downstream innate immune signaling. The manuscript expands on previous work from the authors describing a STING-activation non-canonical autophagy pathway. They show that the activation of STING results in increased K63- and M1- ubiquitin chain formation at LC3B-associated Golgi membranes. Although STING activation induces LC3B lipidation by a non-canonical autophagy pathway, the authors show this pathway has distinct features from the most well-studied non-canonical autophagy pathway, called CASM. Specifically, while both STING and the monensin-induced CASM pathways result in LC3B lipidation as well as increased K63-Ub chain formation, only the STING pathway is able to induce robust linear ubiquitin chain formation. The authors further demonstrate that M1-Ub correlates with the activation of NFkB and IRF3-mediated signaling in THP1 monocytes. Overall the experiments are interesting but preliminary in scope. Importantly, the molecular relationships between STING-induced non-canonical autophagy and linear Ub chain formation remain obscure and the effects of M1-Ub on innate immune signaling are limited to a single cell line. Significant additional work is necessary to support the authors main conclusions that LUBAC synthesis of linear ubiquitin chains regulates STING-mediated innate immune signaling.

1) Although the studies demonstrate differences between STING activation and other non-canonical pathways with regard to the production of M1-Ub chains, additional insight into the mechanisms underlying linear Ub chain formation at LC3 positive membranes is needed. Is LC3B conjugation functionally required for HOIP/LUBAC recruitment to STING foci in response to cGAMP?

This is an excellent suggestion. To rigorously address this issue, we knocked out ATG16L1, an essential component of the LC3B conjugation machinery. Examining the localization of mEGFP-HOIP in ATG16L1KO cells revealed that the lack of LC3B lipidation prevents HOIP localization in perinuclear foci, similarly to Ub, however HOIP still appears in small-dispersed foci (**Fig. 3F,G**). We also now show ATG16L1KO in both HeLa and THP1 cells that LC3B lipidation is not required for M1-Ub chain formation induced by STING (**Fig. 2 and 5**) similar to the effect of SopF in the original manuscript (**now Fig. 3A-C**). We also assessed NFkB activation in the THP1 ATG16L1 KO cells and found no difference from WT cells (**new Fig. 5 and EV5**). Thus, LC3B lipidation is not required for either M1-Ub chain formation or for NFkB activation but may be involved in remodeling membranes where ubiquitin and HOIP reside that changes their spatial distribution. These results more clearly evaluate the relationship between LC3B, M1-Ub chain formation, cellular localization and innate immune signaling by STING.

2) The studies have been almost entirely been conducted in HELA cells expressing STING, except for the one key set of experiments in THP1 looking at immune signaling. Notably, the downstream effects on NFkB target expression are observed in THP1 cells, but not in HELA-STING cells, at least based on the data provided in Supplementary Fig 6. This casts doubt on the broad physiological role of M1-Ub in the control of NFkB and IRF3 signaling, which is the major conclusion of the paper. Additional evidence in other cell lines or tissue types is necessary to more robustly support the conclusions that have been drawn.

We have now expanded our analysis of NFkB signaling in THP1 cells (**Fig. 4 and EV4**) and added results for another cell type, mouse bone marrow derived macrophages (**Fig. EV4**), that also shows HOIP dependent NFkB signaling.

3) For the immunoblotting throughout the studies, the authors show single blots with no attempt as quantification. Quantification of multiple independent experiments with statistical analysis should be provided.

We have now included quantification of several key immunoblotting experiments throughout the manuscript.

4) The physiological relevance of the recruitment of the autophagy cargo receptors TNIP1 and OPTN to LC3 positive membranes during STING activation remains obscure. These studies seem out of place from the rest of the manuscript.

We agree. As this is not clear, other reviewers questioned this aspect, and we have now greatly expanded the manuscript in other directions based on the reviewer comments, we request to delete the TNIP1 and OPTN data.

5) The role of LC3B conjugation in M1-Ub is circumstantial in Fig 5. The role of ATGs required for LC3 conjugation such as ATG7 or ATG3 should be directly tested for their effects on M1-Ub chain formation downstream of cGAMP. Furthermore, as discussed above in point 1, does LC3B play any role in the recruitment of HOIP or other LUBAC components. Overall, these studies lack mechanistic depth.

As described in point 1, we have expanded our findings in ATG16L1KO HeLa and THP1 cells, addressing the requirement for LC3B lipidation in HOIP and ubiquitin spatial localization, M1-Ub chain formation, STING signaling, and innate immune related gene expression at multiple time points.

Referee 3

In the manuscript by Fischer et al., the authors examine the role of LUBAC-mediated M1 linear Ub in STING signalling. They reveal that in addition to K63 Ub (but not K48) activation of STING leads to M1 Ub chain formation associated with LC3B. The authors also demonstrate that STING activation induces the interaction between LC3B and the adaptor proteins, TNIP1 and OPTN, while the deletion of HOIP appears to lead to a loss of STING-dependent cytokine gene expression (in THP-1 cells). Some aspects of STING signalling are not well understood (e.g., NF-kB activation) and here M1 Ub is implicated in STING responses, hence this manuscript is of interest to the field. However, the current data is not always coherent (inconsistency in results from different cell types) and doesn't currently provide enough mechanistic insights into STING signalling other than that M1 Ub is induced in response to STING activation and may be important for cytokine gene expression (in THP1). A major missing piece to this story is also where are the M1 Ub chains are added? Is it directly to STING, or STING complexes, or LC3B alone? This should be examined further.

This manuscript requires major revision before it should be considered further for publication at EMBO J. Below are some specific queries/comments:

1. Introduction: This section should include the findings describing that STING activation also leads to its degradation (which is used as an observation in the manuscript) by the lysosome (see PMID: 29241549) - a process that requires both the ESCRT system and aspects of non-canonical autophagy (see PMID: 36918692 PMID: 36739287 PMID: 37139896). These 3 more recent studies all showed (K63) Ub was also critical for STING degradation and this should be mentioned for the reader and also perhaps written about in the discussion section.

We have included STING lysosomal degradation in the introduction (line 49) and in the discussion (line 318). We also included the role of K63 ubiquitylation of STING in its degradation in the discussion (lines 342-344), citing these sources.

2. Fig.1: A major concern for me was the choice of cell type predominantly used for much of the work. Why did

the authors use HeLa cells expressing an untagged version of STING? not just examine these responses using the endogenous STING levels in HeLa? I believe HeLa express STING at appropriate levels for activation. If not and they are not function with endogenous STING then they are probably not that appropriate for these studies.

We agree and have expanded the number of cell types analyzed. HeLa cells have very low endogenous STING expression compared to immune cell lines, such as THP1 (see Fig. 1 below). Therefore, we generated a HeLa cell line stably expressing STING (HeLa^{STING}) at very low levels that are comparable to THP1 cells in both level of expression and induced gene expression (see below figures 1 and 2). We and others in the field have used this strategy to study certain aspects of STING signaling in HeLa cells (18), such as the spatial localization of STING and other proteins, which is a key type of analysis for our current investigations. HeLa cells are also much better for imaging studies than the small round immune cells in suspension. However, we acknowledge the limitations and concerns of using HeLa cells and that is why we chose to examine a well-established immune cell line with robust STING expression and signaling – THP1 cells. We also have now analyzed and generated HOIP knock down mouse bone marrow derived macrophages (BMDM) cell line. As in THP1 cells, HOIP is required for NFkB activation (Fig. EV4). We now also include data demonstrating that STING signaling (pTBK1, pSTING, pIRF3, and LC3B) in the HeLa^{STING} cell line (Fig. 2 and EV2 and Fig. 4 and EV4) is similar to endogenous STING signaling with both cGAMP and diABZI treatment over time in THP1 cells, supporting the HeLa^{STING} cell line as a valid model system to study certain aspects of STING signaling.

Figure 1. Comparison of STING protein expression detected by immunoblotting in various cell lines.

Figure 2. Comparison of IRF3/Interferon- and NFkB-related gene expression induced by diABZI-mediated STING activation in multiple cell lines.

3. Fig.1: The authors should show the reader a comparison between endogenous STING from WT HeLa vs HeLa-STING, showing at least: STING protein expression levels; and upon activation (i.e., P-STING, P-TBK1, P-IRF3).

As described and shown in the immunoblot above, HeLa cells have very low endogenous STING expression, and it is very difficult to detect STING signaling without overexpression. We have now included new data assessing STING signaling (pTBK1, pSTING, pIRF3, and LC3B) following both cGAMP and diABZI treatment over time in THP1 cells (**Fig. 2 and EV2, Fig. 4 and EV4**), which have robust endogenous STING expression, as well as HeLa^{STING} cells (**Fig. 2 and EV2, Fig. 4 and EV4**).

4. Fig1a: It would be helpful to also include an enlarged triple colour overlay here. Also, what %age of Ub and LC3B localises to STING, both together and also separately (i.e., STING and Ub, STING and LC3B)?

We have now included data detecting the localization of STING relative to LC3B and Ubiquitin foci following both cGAMP and diABZI stimulation of STING (**Fig. 1 and EV1**) with quantification of STING+ and STING+Ub+ LC3B foci, as well as STING+ and STING+LC3B+Ub+ foci (**Fig. 1 and EV1**). These analyses are paired with triple color overlays, per the reviewer's suggestion.

5. Fig.S1d-f: This figure would be improved by also including staining for STING as per Fig.1a.

We have now included representative images and quantification described above for the localization of STING, Ub, and LC3B for both cGAMP and diABZI treatment (**Fig. 1 and EV1**). We have also included data analyzing the co-localization of HOIP and STING at LC3B foci (**Fig. 2 and EV2**). These data all demonstrate the localization of STING at the same LC3B+, Ub+, HOIP+ foci. We do not believe including STING localization in the new Fig. EV1D & E would add information not already shown in these other analyses.

6. Fig. 1e and Fig.S1h: The 2 sentences describing these data should be combined so it is clear which data has used which antibody. Also, some supporting data from more relevant primary immune cells would be important. E.g., does activation of primary human monocytes by STING also lead to a clear increase in detection of M1 Ub chains?

We have removed mention of the second antibody to avoid confusion. Pertaining to the second question, it is technically challenging to KO genes in Human Primary Monocytes, so we tested the role of the M1 Ub by KD of HOIP in BMDM cells. **New Fig. 2** shows time courses of HOIP dependent M1 chain formation in THP1 cells and in BMDM cells. **New Fig. EV4** shows that, as in THP1 cells, HOIP is important for NFkB signaling in BMDM cells.

7. Fig.1f: I found this data a little unclear. It is shown as Fold Change but over what? This should be clear on the y-axis and mentioned in the text. In Fig.1g the axis should also state FC over untreated. Further, why is Fig.1g seemingly presented differently to Fig.1f? This should be consistent: Fig.1g NT and cGAMP - is this 8-fold change statistically significant?

We have changed the axes titles and provided more details in the figure to clarify the analysis.

8. Fig.1: I am left wondering here if STING itself (or the signalling complex) is subject to M1 Ub? It is not clear from the current data but this would be important to examine. For instance, STING IP then IB for M1-Ub (and K63, K48 Ubs to show specificity). This should also be undertaken in THP1 monocytes.

We think this is an interesting suggestion, but a somewhat biased approach and limited in scope. We have related data (in preparation for another manuscript) with K63 ubiquitin chains using a similar approach suggesting that many Golgi proteins including STING are ubiquitylated. Thus, it could be misleading to examine STING alone and draw any conclusions, as activated STING is one of many Golgi localized substrates.

9. Fig.2a: Examines HOIP interactions with LC3B. Does HOIP also appear to interact with STING? Also, what about the other LUBAC components HOIL and SHARPIN - do they interact with LC3B and/or STING?

We think this is beyond the scope of the manuscript, because if we did find any such interactions by co-IP that would open up a whole new aspect to rigorously assess the significance of any such interaction.

10. Fig.2b: Is the remaining band in the blot not HOIP? If so, are these KO or knock down cells?

The reported transcripts of HOIP are 1072 and 921 AA, with calculated kDa of ~120 and ~100 kDa. It could be a second, lower MW isoform. We tested many other antibodies (Fortis Life Sciences A303-560A-T, R&D Systems MAB8039, Abcam ab125189, Aviva Systems Biology ARP43241_P050) – all show various bands around the kDa of HOIP isoforms that appear in different immunoblotting conditions. We cannot confirm whether this is a background band or detection of the second isoform of HOIP.

11. Fig.2c: Here the HOIP, HOIL and Sharpin blots should also be presented below, as in 2b.

We have included HOIP detection in all western blots related to KO or KD of HOIP to show that the lanes are from the KD or KO cells. Since there is little difference in Sharpin or HOIL1 shown now in **Fig. EV2**, we do not feel it would add information to include elsewhere.

12. Fig.2d: Some STING responses are intact in HOIP KO or OTULIN overexpression: In addition, the authors need to examine p-TBK1, p-IRF3 and p-p65 NF-kB. Kinetics should also be examined not just at 8 h. Are these also intact? This should also be done in HOIP KO THP-1 cells later. Also why is the untagged STING not degraded in these cells after 8h?

As requested, we have included new results assessing STING signaling (pTBK1, pSTING, and pIRF3 as well as p-p65 and p-IkBa) at several time points following both cGAMP and diABZI treatment in HeLa^{STING} and THP1 cells (**Fig. 2 and EV2 and Fig. 4 and EV4**). For the second question, the time course of STING detection in THP1 and BMDMs show that endogenous STING is never completely degraded with either cGAMP or diABZI stimulation (**Fig. 2 and EV2**). In HeLa cells overexpressing untagged STING, STING is similarly partially degraded at 8h cGAMP treatment and 4h diABZI treatment (**Fig. EV2**). Additionally, STING is being constitutively expressed by a pSFFV promoter in these cells, therefore it is unlikely complete degradation would be detectable in this system.

13. Fig.2d: Associated to this figure, the data would suggest TBK1 is still actively recruited to STING to mediate STING phosphorylation. Is this the case? Does STING associate with TBK1 via IP?

Absolutely - our data would suggest this, as we show phosphorylation of TBK1 and STING are intact in HOIPKO HeLa and THP1 cells, which many have shown requires the interaction between STING and TBK1 (19–22).

14. Fig.S3e: Here endogenous STING degradation is examined. In these cells the authors should also assess the readouts p-STING, LC3B lipidation, M1 Ub. It would also be informative to show p-TBK1, p-IRF3 and p-p65 NF-kB.

Please see our response to point 2 above, in addition to the accompanying Figures 1 and 2 of this rebuttal. As described above, HeLa cells have very low endogenous STING expression, and it is very difficult to detect STING signaling without overexpression. We have therefore included new data assessing endogenous STING signaling (pTBK1, pSTING, pIRF3, and LC3B) following both cGAMP and diABZI treatment over time in THP1 cell lines, which have robust endogenous STING expression. We believe this is a better model system and clearly demonstrates the effect of HOIPKO on endogenous STING signaling (**new Fig. 2 and EV2 and Fig. 4 and EV4**). We also include new data assessing p65 and IkBa in THP1 and HeLa^{STING} cells over time following both cGAMP and diABZI treatment (**new Fig. 4 and EV4**).

15. Fig.2: Does M1 Ub chain formation by cGAMP require TBK1 i.e. Are M1 Ub chains still induced in TBK1 KO cells (preferably THP1)?

Great question. Another postdoc in the lab is leading a project delving into a large series of TBK1 issues in relationship to STING and ubiquitin.

16. Fig.3a-b: TNIP1 and OPTN are detected at LC3B foci by immunostaining following cGAMP treatment. Here LC3B is overexpressed (as is STING): would this happen under more physiological conditions or endogenous expression levels. Also are TNIP1 and OPTN detected with STING? Or just LC3B?

TNIP1, OPTN, and LC3B antibodies are all raised in Rabbit, thus we are not able to examine endogenous co-localization between these proteins with currently available tools. Considering the other 2 reviewers and the minor conclusions we make in their regard and the large amount of new data added to the manuscript we request to omit these TNIP1 and OPTN data.

17. Fig.3: Is there any functional consequences of TNIP1 and/or OPTN to STING responses? Why are these adaptors localizing to LC3B? What effect on STING responses does KO of these have? Currently this data adds little to the manuscript and shows no relevance to STING immune responses.

We agree that the functional consequences of OPTN and TNIP1 are not known. Thus, we request to remove these data also for reasons mentioned above in response to point 16.

18. Fig.4c: Does loss of HOIP effect STING-induced MAPK or p-p105 NF-kB?

Considering the large body of new results added to the revised version of the manuscript we do not have the capacity to check these additional markers.

19. Fig 4 and Fig.S5: If find the inconsistency in results here very puzzling. The logic that we are following from the beginning is that: STING activation leads to M1 Ub chains, which is dependent on HOIP, which when KO (only in THP1) reduces IFN and NF-kB cytokines as suggested by the title "STING induces LUBAC-mediated synthesis of linear ubiquitin chains to stimulate innate immune signaling". So why are HeLa not the same? I don't believe the throwaway line "however there may be distinct mechanisms in HeLa cells" is an acceptable answer. Together with the overexpression of untagged STING, this left me strongly questioning the relevance of any findings using the HeLa system.

HeLa are not immune cells, that is why we examined immune signaling in THP1 monocytes. We therefore tested the role of the M1 Ub by KD of HOIP in another myeloid cell type, Bone Marrow Derived Macrophages (BMDMs). As in THP1 KO cells, loss of HOIP caused a major reduction of NFkB, corroborating a main conclusion of the manuscript (**Fig. EV4**). Further, during this revision and examining the time course of STING activation in HeLa and THP1 cell lines with both cGAMP and diABZI, per the reviewer's suggestion, we noticed an apparent defect in degradation of overexpressed STING in HOIPKO HeLa^{STING} cells that is more robustly seen at the later time points of cGAMP and diABZI treatment (**Fig. EV2**). While STING induced M1-Ub chain formation is rescued with reconstitution of HOIP in these cells, the observed defect in STING degradation is not (**Fig. 2**). There is also no apparent defect in degradation of the overexpressed STING degradation in the OTULIN overexpressing cell line (**Fig. EV2**), nor in the endogenous STING of the HOIPKO cell line (**Fig. EV2**). These results indicate the defect in overexpressed STING degradation in HOIPKO HeLa cells is not dependent on the loss of M1-Ub chains and may be due to an issue with the subclone selected from the CRISPR KO pool. Further, assessment of IRF3/interferon- and NFkB-mediated gene expression showed opposite results between the 2 methods we used to deplete M1-Ub chains in HeLa cells – HOIPKO increased NFkB gene expression and overexpression of OTULIN decreased NFkB gene expression (**Fig. S1**). A defect in overexpressed STING degradation would explain the results in the HOIPKO HeLa cells, which are skewed higher than WT (**Fig. S1**). Given these differences, and the issues with overexpression the reviewer notes, we are hesitant to rely on results regarding STING signaling from the HOIPKO HeLa cell line overexpressing STING. We are more confident in our results in 2 different myeloid cell lines with endogenous STING that both

show reduced NFkB signaling in either the HOIPKO (pool) or knockdown of HOIP (**Fig. 4 and EV4**). We have revised our presentation of our HeLa data and noted in the text on **lines 119-122 and 174-177** that there may be clonal variations in our HOIPKO and no longer interpret there to be a different mechanism in HeLa cells. We greatly thank the reviewer for bringing this to our attention during revision.

20. Fig.5 lead in: NLRP3 induces pyroptotic cell death.

With apologies - we do not understand how to address this comment.

21. Fig.5a: Does C53 only induce STING activation in HeLa when STING is overexpressed?

We were also surprised and perplexed as to why C53 does not robustly activate STING in THP1 cells. The few studies that have examined C53 have largely used cell lines overexpressing STING (23, 24), with some results in cells with endogenous STING expression (6, 11, 23). It is likely that its agonist activity is more detectable with STING overexpression.

22. Fig.5b: Does C53 induce M1 Ub in WT HeLa with endogenous STING levels?

Please see our response to point 2 above and the accompanying figures. Since there is essentially no detectable IRF3/interferon or NFkB-gene expression in HeLa cells without overexpressing STING (Fig. 2 above), we do not anticipate that STING-mediated M1 Ub chain formation is detectable in these cells with any STING agonist.

23. Fig. S8e: In THP1 C53 doesn't induce STING activation but also appears not to induce significant LC3B lipidation. Why? The data is hard to interpret with such poor induction of LC3B lipidation.

C53 is a reported STING agonist that also blocks LC3B lipidation, so it is not expected to induce LC3B lipidation (6, 11).

24. Fig.5c-d: Why do the ISGs IFIT3 and ISG15 go up with C53 + diABZI treatment? If IFN β goes down, shouldn't they also be reduced?

This is a good point. As this reviewer already points out, there is complex cross talk between NFkB and IFN responses and this is one such example.

25. Related to Fig.e-f: What is the effect of C53 on STING signalling generally in THP1? The authors should examine p-TBK1, p-IRF3 and p-p65 NF-kB as well.

We examined C53 activation of STING in cell lines with robust endogenous STING (THP1 and U937) over time to ensure we are not missing signaling at different time points (see Figure 3 below). We detected very little phosphorylation of TBK1, IRF3, and STING in both cell lines, with the highest signaling around 1-2 hours.

Figure 3. Comparison of STING signaling with C53 treatment over time in U937 and THP1 cells.

26. Fig.5f: Does SopF reduce STING-induced LC3B lipidation examined by WB? If SopF blocks LC3B lipidation generally, shouldn't M1 Ub after cGAMP be reduced in the presence of SopF not unchanged? I find the logic here very unclear. SopF experiments should also be conducted in THP1 cells.

We published SopF expression blocks STING-mediated LC3B lipidation examined by immunoblotting in Fischer et al., 2020. This has been corroborated by others (25, 26). Here we tested the hypothesis - if LC3B lipidation is required for M1 ubiquitin chain formation, then SopF would also block M1 ubiquitin chain formation - and we show that SopF does not block M1 ubiquitin chain formation in new **Figure 3**. We now corroborate this conclusion by showing ATG16L1 KO also does not inhibit M1 ubiquitin chain formation in both HeLa (**Fig. 3**) and THP1 cells (**Fig. 5**).

27. Fig.5 conclusion: "suggests that STING activation induced LC3B lipidation may promote ubiquitin chain formation" but its stated SopF blocks LC3B lipidation. As stated above why doesn't SopF therefore reduce M1 Ub?

Thank you for this point. Considering our new, expanded investigation into the relationship between LC3B lipidation and M1-Ub chain formation, we have removed this statement and modified our conclusions.

28. Discussion: IFN is also affected by HOIP KO. In some cell types NF-kB contributes to a full IFN β transcriptional response - see the IFN β enhanceosome. Hence, without examining p-IRF3 in these studies one cannot conclude where this M1 Ub may be affecting things? Is it just acting on NF-kB?

We have included data assessing pIRF3 following both cGAMP and diABZI treatment over time in WT and HOIPKO THP1 and WT and HOIPKO HeLa^{STING} cells (**Fig. 4 and EV4**). Our results indicate that HOIP is dispensable for pIRF3, suggesting HOIP is involved specifically in NFkB signaling and may regulate IFN β through the enhanceosome as the reviewer suggests.

29. Discussion: The mention that TBK1 recruitment to STING may involve OPTN should either be investigated or not mentioned.

Agreed, we will delete this comment.

30. General concern: The authors use either 60 or (more often) 120 $\mu\text{g}/\text{mL}$ of 2'3'-cGAMP - which are very high concentrations (commonly used between 5-20 $\mu\text{g}/\text{mL}$). It is currently unclear exactly how cells are stimulated with cGAMP. Please explain how this is done. Is cGAMP added to media or transfected? This form of cGAMP is generally transfected while other stabilised forms of 2'3'-cGAMP e.g. 2'3'-cGAMP(PS)₂ (Rp/Sp) are added to media at much lower concentrations. Otherwise for human cells one can also use diABZI (as is done

in some later aspects of this manuscript. Also, why is diABZI used for THP1 experiments and cGAMP for HeLa? Does diABZI elicit the same results in HeLa and vice versa?

We use a high concentration because, unlike in some other studies, we do not transfect cGAMP into cells, we simply add it to the media. We clarified our treatment conditions in the Materials and Methods. We showed a dose response curve to cGAMP treatment in our previous paper, Fischer et al., 2020. We have now added time courses for both cGAMP and diABZI for key experiments in HeLa and in THP1 cells. diABZI and cGAMP elicited the same results with different dynamics in both cell lines.

31. Authors state that M1 Ub may be associated with STING-induced LC3B: Use C53 to show no STING activation but no M1 Ub?

Exactly, we show the effect of C53 on M1 Ub in **Figure EV3**.

References

1. M. L. Matsumoto, K. C. Dong, C. Yu, L. Phu, X. Gao, R. N. Hannoush, S. G. Hymowitz, D. S. Kirkpatrick, V. M. Dixit, R. F. Kelley, Engineering and Structural Characterization of a Linear Polyubiquitin-Specific Antibody. *Journal of Molecular Biology* **418**, 134–144 (2012).
2. K. Newton, M. L. Matsumoto, R. E. Ferrando, K. E. Wickliffe, M. Rape, R. F. Kelley, V. M. Dixit, “Using Linkage-Specific Monoclonal Antibodies to Analyze Cellular Ubiquitylation” in *Ubiquitin Family Modifiers and the Proteasome: Reviews and Protocols*, R. J. Dohmen, M. Scheffner, Eds. (Humana Press, Totowa, NJ, 2012; https://doi.org/10.1007/978-1-61779-474-2_13) *Methods in Molecular Biology*, pp. 185–196.
3. H. Ishikawa, G. N. Barber, STING is an endoplasmic reticulum adaptor that facilitates innate immune signalling. *Nature* **455**, 674–678 (2008).
4. T. Saitoh, N. Fujita, T. Hayashi, K. Takahara, T. Satoh, H. Lee, K. Matsunaga, S. Kageyama, H. Omori, T. Noda, N. Yamamoto, T. Kawai, K. Ishii, O. Takeuchi, T. Yoshimori, S. Akira, Atg9a controls dsDNA-driven dynamic translocation of STING and the innate immune response. *Proc Natl Acad Sci U S A* **106**, 20842–6 (2009).
5. K. Mukai, H. Konno, T. Akiba, T. Uemura, S. Waguri, T. Kobayashi, G. N. Barber, H. Arai, T. Taguchi, Activation of STING requires palmitoylation at the Golgi. *Nat Commun* **7**, 11932 (2016).
6. J. Xun, Z. Zhang, B. Lyu, D. Lu, H. Yang, G. Shang, J. X. Tan, A conserved ion channel function of STING mediates non-canonical autophagy and cell death. bioRxiv [Preprint] (2023). <https://doi.org/10.1101/2023.08.26.554976>.
7. H. Kemmoku, K. Takahashi, K. Mukai, T. Mori, K. M. Hirosawa, F. Kiku, Y. Uchida, Y. Kuchitsu, Y. Nishioka, M. Sawa, T. Kishimoto, K. Tanaka, Y. Yokota, H. Arai, K. G. N. Suzuki, T. Taguchi, Single-molecule localization microscopy reveals STING clustering at the trans-Golgi network through palmitoylation-dependent accumulation of cholesterol. *Nat Commun* **15**, 220 (2024).
8. X. Tu, T.-T. Chu, D. Jeltema, K. Abbott, K. Yang, C. Xing, J. Han, N. Dobbs, N. Yan, Interruption of post-Golgi STING trafficking activates tonic interferon signaling. *Nat Commun* **13**, 6977 (2022).
9. M. Gentili, B. Liu, M. Papanastasiou, D. Dele-Oni, M. A. Schwartz, R. J. Carlson, A. M. Al'Khafaji, K. Krug, A. Brown, J. G. Doench, S. A. Carr, N. Hacohen, ESCRT-dependent STING degradation inhibits steady-state and cGAMP-induced signalling. *Nat Commun* **14**, 611 (2023).
10. T. D. Fischer, C. Wang, B. S. Padman, M. Lazarou, R. J. Youle, STING induces LC3B lipidation onto single-membrane vesicles via the V-ATPase and ATG16L1-WD40 domain. *J Cell Biol* **219** (2020).

11. B. Liu, R. J. Carlson, I. S. Pires, M. Gentili, E. Feng, Q. Hellier, M. A. Schwartz, P. C. Blainey, D. J. Irvine, N. Hacohen, Human STING is a proton channel. *Science* **381**, 508–514 (2023).
12. C. H. Emmerich, A. Ordureau, S. Strickson, J. S. C. Arthur, P. G. A. Pedrioli, D. Komander, P. Cohen, Activation of the canonical IKK complex by K63/M1-linked hybrid ubiquitin chains. *Proceedings of the National Academy of Sciences* **110**, 15247–15252 (2013).
13. T. L. Haas, C. H. Emmerich, B. Gerlach, A. C. Schmukle, S. M. Cordier, E. Rieser, R. Feltham, J. Vince, U. Warnken, T. Wenger, R. Koschny, D. Komander, J. Silke, H. Walczak, Recruitment of the Linear Ubiquitin Chain Assembly Complex Stabilizes the TNF-R1 Signaling Complex and Is Required for TNF-Mediated Gene Induction. *Molecular Cell* **36**, 831–844 (2009).
14. S. Rahighi, F. Ikeda, M. Kawasaki, M. Akutsu, N. Suzuki, R. Kato, T. Kensche, T. Uejima, S. Bloor, D. Komander, F. Randow, S. Wakatsuki, I. Dikic, Specific Recognition of Linear Ubiquitin Chains by NEMO Is Important for NF- κ B Activation. *Cell* **136**, 1098–1109 (2009).
15. J. M. Ramanjulu, G. S. Pesiridis, J. Yang, N. Concha, R. Singhaus, S.-Y. Zhang, J.-L. Tran, P. Moore, S. Lehmann, H. C. Eberl, M. Muelbauer, J. L. Schneck, J. Clemens, M. Adam, J. Mehlmann, J. Romano, A. Morales, J. Kang, L. Leister, T. L. Graybill, A. K. Charnley, G. Ye, N. Nevins, K. Behnia, A. I. Wolf, V. Kasparcova, K. Nurse, L. Wang, A. C. Puhl, Y. Li, M. Klein, C. B. Hopson, J. Guss, M. Bantscheff, G. Bergamini, M. A. Reilly, Y. Lian, K. J. Duffy, J. Adams, K. P. Foley, P. J. Gough, R. W. Marquis, J. Smothers, A. Hoos, J. Bertin, Design of amidobenzimidazole STING receptor agonists with systemic activity. *Nature* **564**, 439–443 (2018).
16. R. D. Luteijn, S. A. Zaver, B. G. Gowen, S. K. Wyman, N. E. Garelis, L. Onia, S. M. McWhirter, G. E. Katibah, J. E. Corn, J. J. Woodward, D. H. Raulet, SLC19A1 transports immunoreactive cyclic dinucleotides. *Nature* **573**, 434–438 (2019).
17. C. Ritchie, A. F. Cordova, G. T. Hess, M. C. Bassik, L. Li, SLC19A1 Is an Importer of the Immunotransmitter cGAMP. *Mol Cell* **75**, 372-381.e5 (2019).
18. Y. Liu, P. Xu, S. Rivara, C. Liu, J. Ricci, X. Ren, J. H. Hurley, A. Ablasser, Clathrin-associated AP-1 controls termination of STING signalling. *Nature* **610**, 761–767 (2022).
19. C. Zhang, G. Shang, X. Gui, X. Zhang, X. C. Bai, Z. J. Chen, Structural basis of STING binding with and phosphorylation by TBK1. *Nature* **567**, 394–398 (2019).
20. S. Yum, M. Li, Y. Fang, Z. J. Chen, TBK1 recruitment to STING activates both IRF3 and NF- κ B that mediate immune defense against tumors and viral infections. *Proc Natl Acad Sci U S A* **118** (2021).
21. Y. Tanaka, Z. J. Chen, STING specifies IRF3 phosphorylation by TBK1 in the cytosolic DNA signaling pathway. *Sci Signal* **5**, ra20 (2012).
22. S. Liu, X. Cai, J. Wu, Q. Cong, X. Chen, T. Li, F. Du, J. Ren, Y.-T. Wu, N. V. Grishin, Z. J. Chen, Phosphorylation of innate immune adaptor proteins MAVS, STING, and TRIF induces IRF3 activation. *Science* **347** (2015).
23. D. C. Pryde, S. Middy, M. Banerjee, R. Shrivastava, S. Basu, R. Ghosh, D. B. Yadav, A. Surya, The discovery of potent small molecule activators of human STING. *Eur J Med Chem* **209**, 112869 (2021).
24. D. Lu, G. Shang, J. Li, Y. Lu, X. Bai, X. Zhang, Activation of STING by targeting a pocket in the transmembrane domain. *Nature* **604**, 557–562 (2022).
25. Y. Xu, S. Cheng, H. Zeng, P. Zhou, Y. Ma, L. Li, X. Liu, F. Shao, J. Ding, ARF GTPases activate Salmonella effector SopF to ADP-ribosylate host V-ATPase and inhibit endomembrane damage-induced autophagy. *Nat Struct Mol Biol* **29**, 67–77 (2022).

26. K. M. Hooper, E. Jacquin, T. Li, J. M. Goodwin, J. H. Brumell, J. Durgan, O. Florey, V-ATPase is a universal regulator of LC3-associated phagocytosis and non-canonical autophagy. *J Cell Biol* **221**, e202105112 (2022).

Dear Richard,

Thank you for submitting the revised version of your manuscript, which addresses the concerns of the referees. This revised version has now been re-reviewed; I attach the second referee reports to the bottom of this mail. As you will see, you have addressed the referees' concerns to their satisfaction. Before I can finally accept the manuscript, there are some remaining editorial points which need to be addressed. In this regard, would you please:

- rename the "Data and Materials Availability" section the "Data Availability",
- remove the author credit section from the manuscript,
- include callouts in the main manuscript text for Fig. 3D-E, 4E-H,
- upload figures as individual, high-resolution figure files,
- add the manuscript title to appendix title page (Appendix for STING induces HOIP-mediated synthesis of M1 ubiquitin chains to stimulate NF κ B signaling); add page numbers to the appendix table of contents using the nomenclature "Appendix Figure S1-S2",
- remove the "Instructions" section from the Reagents and Tools table and remove the Reagents and Tools table from the manuscript file, uploading it as an individual file,
- save Source Data in a scheme of one figure/folder and upload as .zip files. E.g. all the Source data files for figure 1 need to be saved in a single folder and this needs to be zipped and then uploaded as "SD figure 1.zip" file. For EV and/or appendix figures, ZIP together all source data,
- ensure dataset MSV000092940 is publicly available,
- provide figure titles for figures EV 1-5 in the manuscript,
- state exact p values in the legends of figures 4e-h; 5b; EV 4b-e; EV 5c-d, correct a mismatch between the annotated p values in the figure legend and the annotated p values in the figure file in figures 4e-h; 5b; EV 4d-e; EV 5c-d,
- define the measure of centre and error bars in the legends of figures 1b, h; 2b; 4e-h; 5b-c; EV 1f; EV 4b-e; and EV 5c-d,
- rename the movie file "Movie EV1" with the corresponding callout in the manuscript, remove the legend from the manuscript file and zip it with the movie file, and
- correct the section order as follows: title page with complete author information, abstract, keywords, introduction, results, discussion, methods, data availability section, acknowledgements, disclosure and competing interests statement, references, main figure legends, tables, expanded figure legends.

We include a synopsis of the paper on our website (see <http://emboj.emboPress.org/>). Please provide me with a general summary image, a two-sentence summary statement and 3-5 bullet points that capture the key findings of the paper.

I look forward to receiving these changes. EMBO Press is an editorially independent publishing platform for the development of EMBO scientific publications.

Best wishes,

William

William Teale, PhD
Editor
The EMBO Journal
w.teale@embojournal.org

See also figure legend guidelines: <https://www.emboPress.org/page/journal/14602075/authorguide#figureformat>

- a point-by-point response to the referees' comments, with a detailed description of the changes made (as a word file).
- a word file of the manuscript text.
- individual production quality figure files (one file per figure)
- a complete author checklist, which you can download from our author guidelines (<https://www.emboPress.org/page/journal/14602075/authorguide>).

- Expanded View files (replacing Supplementary Information)

We realize that it is difficult to revise to a specific deadline. In the interest of protecting the conceptual advance provided by the work, we recommend a revision within 3 months (15th Jan 2025). Please discuss the revision progress ahead of this time with the editor if you require more time to complete the revisions. Use the link below to submit your revision:

Referee #1:

The authors have done extensive revisions and responded to raised questions. The manuscript is now acceptable for publication.

Referee #3:

I commend the authors on their hard work in addressing the majority of all the reviewers' comments. In particular, I appreciate the undertaking on many new experiments in THP-1 cells that adds much to the immunological relevance of the findings.

I am satisfied that my concerns have been largely answered and feel the manuscript should now be accepted.

One minor point that requires revision is the new iBMDM in figure EV4 D-E. The gene names should be changed to reflect murine cells.

All editorial and formatting issues were resolved by the authors.

Dear Richard,

I am pleased to inform you that your manuscript has been accepted for publication in the EMBO Journal.

Congratulations to you and Tara on a really exciting study!

Best wishes,

William

William Teale, PhD
Editor
The EMBO Journal
w.teale@embojournal.org
